# Endogenous corazonin signaling modulates the post-mating switch in behavior and physiology in females of the brown planthopper and *Drosophila*

**Ning Zhang[1], Shao-Cong Su[1], Ruo-Tong Bu[1], Yijie Zhang[1], Lei Yang[1], Jie Chen[1], Dick R Nässel[2], Congfen Gao[1], Shun-Fan Wu[1]\***

[1]Sanya Institute of Nanjing Agricultural University, College of Plant Protection and College of Sciences, State Key Laboratory of Agricultural and Forestry Biosecurity, Nanjing Agricultural University, Nanjing, China; [2]Department of Zoology, Stockholm University, Stockholm, Sweden

## eLife Assessment

This **important** study presents **convincing** evidence that uncovers a novel signaling axis impacting the post-mating response in females of the brown planthopper. The findings open several avenues for testing the molecular and neurobiological mechanisms of mating behavior in insects, and in the revised version the authors provide further evidence supporting their conclusions.

**\*For correspondence:**
wusf@njau.edu.cn

**Competing interest:** The authors declare that no competing interests exist.

**Abstract** Mating in insects typically triggers a post-mating response (PMR) in females, characterized by reduced receptivity to re-mating and increased oviposition, which ensures numerous and viable offspring and male paternity. This PMR is induced by male seminal factors, such as sex peptide in *Drosophila melanogaster*, as well as intrinsic female signaling components. The latter signaling remains poorly understood in most insects, including the devastating rice pest, the brown planthopper (BPH) *Nilaparvata lugens*. Here, we show that the neuropeptide corazonin (CRZ) and its receptor (CrzR) are critical for the PMR in female BPHs. Peptide injection, RNAi knockdown, and CRISPR/Cas9 mutagenesis confirm that intact CRZ signaling reduces re-mating frequency and increases ovulation in mated BPH females. The CrzR is highly expressed in the female reproductive tract, and CrzR knockdown phenocopies CRZ diminishment. Importantly, female CRZ/CrzR signaling is required for male seminal factors, such as the peptide maccessin, to induce the PMR; with disrupted *CrzR* signaling, injection of seminal fluid or maccessin fails to reduce female receptivity. Notably, CRZ is not produced in male accessory glands (MAGs) of BPHs and thus not transferred during copulation. We furthermore demonstrate that also in *D. melanogaster* disrupted CRZ signaling increases female re-mating and reduces oviposition, while CRZ injection suppresses virgin receptivity and increases oviposition. Finally, we detected no CRZ in the MAG of *D. melanogaster*, supporting its role as an endogenous signal in the female PMR also in this species. In summary, our findings reveal a conserved role of endogenous CRZ signaling in regulating the female PMR and demonstrate that female CRZ signaling and male-derived signals cooperate to induce post-mating transitions in BPHs and *D. melanogaster*. CRZ is a paralog of the peptide gonadotropin-releasing hormone, known to regulate reproduction in vertebrates, including humans, suggesting evolutionary conservation of an ancient function.

## Introduction

Reproductive behavior in animals is influenced by a complex interplay between external sensory inputs and internal states, and is regulated by neuronal and hormonal signaling. In insects, males typically initiate mating through courtship, while females determine acceptance by integrating external sensory cues with their internal state. Consequently, female reproductive activity is modulated by multiple signaling pathways (*Kim et al., 2024*; *Sun et al., 2023b*; *Wang et al., 2020*; *Wang et al., 2021*; *Kubli, 2003*; *Yapici et al., 2008*; *Yang et al., 2009*; *Rezával et al., 2012*). Importantly, in insects, successful copulation commonly induces profound physiological and behavioral changes in the females (*Kubli, 2003*; *Camus et al., 2018*; *Craig, 1967*; *Judson, 1967*; *Liu et al., 2024*; *Liu and Kubli, 2003*; *Chapman et al., 2003*; *Duvall et al., 2017*; *Wolfner, 2002*; *Gorter et al., 2016*; *Chen et al., 1988*).

Following their first mating, insect females typically reject further courtship attempts and initiate oviposition. This distinct shift in behavior and physiology is referred to as a post-mating response (PMR) (*Kubli, 2003*; *Yapici et al., 2008*; *Yang et al., 2009*; *Ellendersen and von Philipsborn, 2017*; *Klowden, 1999*; *Manning, 1962*; *McCall, 2000*; *Lung et al., 2002*). Induction of a PMR has been documented across diverse insect taxa, for example, in *D. melanogaster*, mosquitoes *Anopheles gambiae, Aedes aegypti,* crickets (*Gryllodes sigillatus*), and the brown planthopper (BPH), *Nilaparvata lugens* (*Yapici et al., 2008*; *Craig, 1967*; *Duvall et al., 2017*; *Chen et al., 1988*; *Manning, 1962*; *Zhang et al., 2025*; *Rines et al., 2023*; *Tripet et al., 2003*; *Klowden and Russell, 2004*; *Naccarati et al., 2012*).

Induction of a PMR commonly requires the transfer of specific seminal fluid components during copulation. In *D. melanogaster*, seminal fluid peptides, including sex peptide (SP) and DUP99B produced in male accessory glands (MAG), enter the female reproductive tract, bind to specific receptors on sensory neurons, and activate signaling pathways that drive post-mating behavior and physiology (*Yapici et al., 2008*; *Liu and Kubli, 2003*; *Chapman et al., 2003*; *Chen et al., 1988*; *Saudan et al., 2002*). Additional factors such as Acp26Ab and ovulin (Acp26Aa) are also transferred with the seminal fluid and modulate female reproductive physiology in *D. melanogaster* (*Herndon and Wolfner, 1995*; *Ram and Wolfner, 2007*; *Heifetz et al., 2005*; *Gligorov et al., 2013*). Notably, it has been shown that the PMR-inducing substances, produced in the MAG of various insects, are highly species-specific and not even conserved across related taxonomic groups (*Zhang et al., 2025*; *Hopkins et al., 2024*).

While MAG-derived sex peptide (SP) and its receptor (SPR) are key regulators of the female post-mating response (PMR) in *D. melanogaster* (*Kubli, 2003*; *Yapici et al., 2008*; *Yang et al., 2009*; *Wolfner, 2002*), additional signaling pathways intrinsic to females also modulate this process. Interneurons within abdominal neuromeres of the ventral nerve cord (VNC) promote receptivity to males by signaling via myoinhibitory peptide (MIP) to brain circuits (*Jang et al., 2017*). Furthermore, the neuropeptide diuretic hormone 44 (DH44) modulates female sexual receptivity through a sex-specific brain circuit; mating suppresses DH44 signaling, thereby reducing receptivity (*Kim et al., 2024*). During sexual maturation, leucokinin (LK)-expressing neurons (ABLKs) in the abdominal ganglion, downstream of SP-activated circuitry, suppress female receptivity (*Chen et al., 2025*). In addition to changes in receptivity, the *D. melanogaster* PMR includes a shift in diet preference toward yeast consumption. This altered feeding behavior is mediated by specific peptidergic neurons (ALKs) in the brain through LK signaling (*Liu et al., 2024*).

Although the mechanisms and neuronal pathways underlying the (PMR) in *D. melanogaster* are well characterized (*Wang et al., 2020*; *Wang et al., 2021*; *Yapici et al., 2008*; *Yang et al., 2009*; *Rezával et al., 2012*; *Avila et al., 2011*; *Häsemeyer et al., 2009*; *Auer and Benton, 2016*), this process remains poorly understood in most non-model insects. A prominent example is the brown planthopper (BPH), *N. lugens* (Hemiptera). As a monophagous rice pest, the BPH exhibits robust reproductive capacity and remarkable environmental adaptability (*Lin et al., 2025*). It has also developed high resistance to diverse chemical pesticides (*Wu et al., 2018*; *Song et al., 2022*; *Ye et al., 2024*). Consequently, elucidating reproductive physiology and behavior in BPHs is crucial for developing novel control strategies.

In BPHs, the recently identified MAG-derived peptide maccessin (macc) is transferred during copulation and triggers a PMR in females, although its receptor remains unknown (*Zhang et al., 2025*). Notably, the same study demonstrated that a second peptide, ion transport-like peptide 1 (ITPL-1),

also modulates the PMR. Whereas macc is male-specific, ITPL-1 is produced both in the MAG and endogenously in females, where it reduces female receptivity to courting males (*Zhang et al., 2025*). Given its endogenous female expression, ITPL-1 likely functions as an intrinsic signaling component in the PMR of BPHs, analogous to the role of myoinhibitory peptide (MIP) and DH44 in *D. melanogaster* females (*Kim et al., 2024*; *Jang et al., 2017*).

Here, we explored the role of another neuropeptide, corazonin (CRZ), in the reproductive behavior of BPHs. In *D. melanogaster* males, CRZ is released from interneurons of the ventral nerve cord (VNC) and acts on serotonergic projection neurons to regulate ejaculation and copulation duration (*Tayler et al., 2012*), mitigates mating-induced heart rate acceleration for physiological recovery (*Li et al., 2025*) and influences post-mating dietary preference (*Liu et al., 2024*). Surprisingly, the role of CRZ signaling in female reproduction remains unexplored. To address this gap, we investigated the role of CRZ in female reproduction, including the PMR, in BPHs and *D. melanogaster*.

Here, we demonstrate that CRZ induces a post-mating switch in receptivity in females of both species. CRZ injection in virgin females reduces receptivity to courting males and increases oviposition, while *Crz* knockdown or knockout in mated females restores receptivity to re-mating and decreases oviposition. Notably, CRZ is not produced in the male accessory gland (MAG) of either species and thus the peptide is not transferred during copulation, but acting as an endogenous female peptide signal. In BPHs, the CRZ receptor (CrzR) is highly expressed in the female reproductive tract and both CRISPR-Cas9-mediated *CrzR* knockout and RNAi-mediated knockdown disrupt the PMR, phenocopying CRZ deficiency. Critically, receptivity changes in virgin BPH female, whether induced by MAG extract (containing seminal fluid proteins) or CRZ injection, depend on the presence of the CrzR. Collectively, our data demonstrate that endogenous female CRZ signaling and male seminal cues are both required to coordinate post-mating changes in physiology and behavior in females of both insects. CRZ is a paralog of the peptide gonadotropin-releasing hormone, known to regulate reproduction in vertebrates, including humans, suggesting evolutionary conservation of an ancient function.

## Results

### CRZ signaling reduces receptivity to courting males and stimulates oviposition in *N. lugens* females

While numerous studies have established roles of the neuropeptide CRZ in regulating aspects of male reproduction across diverse insect species (*Tayler et al., 2012*; *Gospocic et al., 2017*; *Hou et al., 2018*; *Shohat-Ophir et al., 2012*; *Zer-Krispil et al., 2018*; *Cheng et al., 2022*; *Yang et al., 2008*), research addressing its potential functions in female reproduction remains conspicuously lacking.

This study investigates the role of CRZ in female reproduction, starting with the brown planthopper (BPH), *Nilaparvata lugens*, a major agricultural pest. Initially, the *Crz* gene (GenBank accession: AB817247.1) was cloned based on transcriptome data (*Tanaka et al., 2014*). The deduced mature CRZ sequence in BPH, pQTFQYSRGWTNamide, is identical to that reported for most studied insect species (*Figure 1—figure supplement 1A*). Consistent with other known CRZ precursors, the BPH precursor encodes a single CRZ peptide starting at the first amino acid of the propeptide (*Figure 1—figure supplement 1A and B*). A comparison of the organization of selected CRZ precursor genes is shown in *Figure 1—figure supplement 1B*. Genomic analysis revealed that the CRZ precursor is encoded by two exons separated by a single intron (*Figure 1—figure supplement 1B*). The mature CRZ peptide and a scrambled control peptide (sCRZ) were synthesized for pharmacological experiments.

First, we investigated the role of CRZ signaling in the reproductive behavior and PMR of female BPHs. The PMR in BPHs, is characterized by reduced receptivity to courting males, including display of rejection behaviors, and increased oviposition (*Zhang et al., 2025*). We found that injection of CRZ significantly reduced receptivity of virgin female BPHs to courting males (*Figure 1A and B*). Furthermore, virgin females injected with CRZ actively rejected courting males, exhibiting a distinct rejection behavior characterized by ovipositor extrusion and kicking or fleeing. This behavior was not observed in control females injected with sCRZ (*Figure 1—video 1*). Notably, the CRZ-induced mating refusal behavior in virgin females was no longer observed 24 hr after injection (*Figure 1—figure supplement 2A*). While injection of CRZ peptide did not stimulate egg production in virgin BPHs (*Figure 1—figure supplement 2B*), it significantly promoted oviposition in mated females (*Figure 1C*).

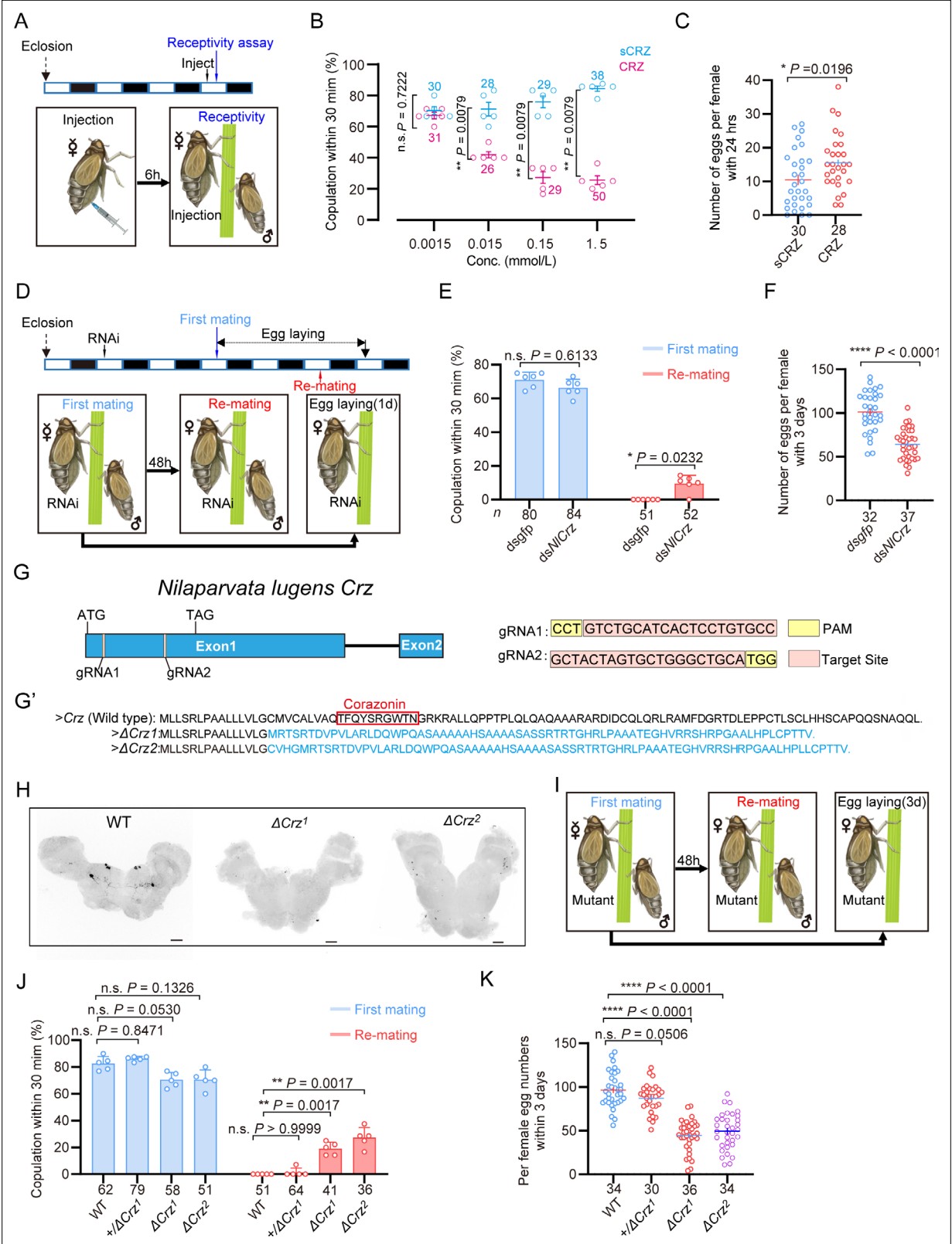

**Figure 1.** Corazonin (CRZ) signaling affects the post-mating response (PMR) and oviposition in female *N. lugens*. Wild-type males were used in all behavioral assays. (**A**) Experimental design for panels **B-C**. The white/black segments on the experimental time lines denote light/dark periods, respectively. (**B**) Receptivity of virgin females 6 hr post-injection, using different doses of CRZ (sCRZ as control), each virgin female BPH was injected with a calibrated glass capillary needle directly into the abdomen with 100 ng/10 ng/1 ng/0.1 ng of peptide dissolved in 50 nL of 1x PBS balanced salt

*Figure 1 continued*

solution. The graph shows the percentage of females copulating within 30 min. **$p<0.01$ vs. control; ns (non-significant): $p>0.05$ (Mann-Whitney test). The small circles denote the number of replicates, ≥5 insects/replicate; the numbers above the curves denote total number of animals. (**C**) Eggs laid per mated female 24hr post-copulation. We used the highest CRZ dose from panel B (sCRZ, control). Each mated female BPH was injected with a calibrated glass capillary needle directly into the abdomen with 100 ng of peptide dissolved in 50 nL of 1x PBS balanced salt solution. *$p<0.05$ (Student's t-test). Data: mean ± s.e.m (≥4 biological replicates, ≥6 insects/replicate). The numbers below the bars denote total number of animals. (**D**) Experimental design for panels **E**-**F**. (**E**) Receptivity of virgin (first mating) and mated females after *Crz*-RNAi (*dsgfp* as control). Graph displays the percentage of females copulating within 30 min. The small circles denote the number of replicates, ≥7 insects/replicate; the numbers below the bars denote total number of animals. Statistical comparisons of proportions were performed using chi-square tests for contingency tables, $p<0.05$ was considered significant. (**F**) Total number of eggs laid per female over 72 hr after *Crz*-RNAi (*dsgfp* as control). ****$p<0.0001$ (Student's *t*-test). The small circles and the numbers below the bars denote total number of animals, (≥4 biological replicates, ≥8 insects/replicate). Data are shown as mean ± s.e.m. (≥3 biological replicates, ≥8 insects/replicate). (**G**) Schematic diagram of the *Crz* gene and sgRNA (single-guide RNA) design for *Crz* mutant production. The *Crz* gene consists of 2 exons, and ATG and TAG are located on Exon 1. The target site of the 20 bp sgRNA on Exon 1 is highlighted in pink. The PAM sites are indicated in yellow. The two knockout strains *ΔCrz¹* and *ΔCrz²* were unable to produce mature CRZ peptides. (**G'**) The amino acid sequences obtained by translating the mutated base sequences. (**H**) Absence of CRZ immunoreactivity in homozygous *ΔCrz¹* and *ΔCrz²* mutants compared to wild-type (WT) expression. Scale bars: 50 μm. (**I**) Experimental design for panels **J**-**K**. (**J**) Receptivity of virgin/mated females across genotypes. No effect was seen on virgin receptivity, only the re-mating was affected. The small circles denote the number of replicates; ≥6 insects/replicate, the numbers below the bars denote total number of animals. Statistical comparisons of proportions were performed using chi-square tests for contingency tables, $p<0.05$ was considered significant. (**K**) Eggs laid 72 hr post-mating. The numbers below the bars denote total number of animals, ≥4 biological replicates, ≥8 insects/replicate. Data are shown as mean ± s.e.m. ****$p<0.0001$; Student's *t*-test.

The online version of this article includes the following video, source data, and figure supplement(s) for figure 1:

**Source data 1.** Source data for all graphs presented in *Figure 1*.

**Figure supplement 1.** Evolutionary conservation of corazonin (CRZ) peptide sequences in *Nilaparvata lugens* and other arthropods.

**Figure supplement 2.** The corazonin (*Crz*) gene-silencing efficacy in female insects following dsRNA injection assayed by qPCR.

**Figure supplement 2—source data 1.** Source data for all graphs presented in *Figure 1—figure supplement 2*.

**Figure supplement 3.** Phenotypic effects of corazonin (CRZ) peptide injection.

**Figure supplement 4.** Characterization of corazonin (*Crz*) mutants in *N. lugens*.

**Figure 1—video 1.** Virgin brown planthopper females reject mating after CRZ peptide injection.
https://elifesciences.org/articles/109297/figures#fig1video1

To further investigate CRZ action, we performed RNA interference (RNAi) by injecting double-stranded *Crz* RNA (*dsNlCrz*) into female BPHs, which efficiently reduced *Crz* expression (*Figure 1—figure supplement 2*). In *N. lugens*, female receptivity is transiently suppressed after mating and does not begin to recover until at least 72 hr after the first mating (*Zhang et al., 2025*); therefore, remating assays were performed at 48 hr post-mating as shown in *Figure 1D*. Knockdown of *Crz* resulted in increased receptivity with 10% of the mated females mating again, and significantly reduced oviposition to approximately 60% of control levels (*Figure 1E and F*). In contrast, *Crz* knockdown had no effect on the receptivity of virgin females (*Figure 1E*).

Since RNAi-mediated knockdown of *Crz* in female BPHs left a residual level of gene expression (*Figure 1—figure supplement 2*), we generated two *Crz* deletion alleles, *ΔCrz¹* and *ΔCrz²*, using CRISPR/Cas9 (*Figure 1G–G'*) to achieve a complete ablation. Both alleles harbor deletions eliminating the mature CRZ peptide coding sequence (*Figure 1G"*). Sanger sequencing confirmed that the deletions compromised essential triplet codons within the *Crz* open reading frame (*Figure 1—figure supplement 4*), thereby efficiently disrupting CRZ peptide expression (*Figure 1G" and H*). Immunohistochemical analysis further validated the mutants, revealing a complete absence of anti-CRZ immunolabeling in homozygous *ΔCrz¹* and *ΔCrz²* females (*Figure 1H*). Homozygous *ΔCrz¹* and *ΔCrz²* virgin females are fully viable and fertile. They exhibit a courtship vigor similar to controls and their initial mating success rate with wild-type males exceeds 60% (*Figure 1I and J*). However, 25% of the mated mutant females were receptive to further mating (*Figure 1J*), and oviposition was significantly reduced to approximately 50% of the wild-type levels (*Figure 1K*). To further test the *Crz* mutants, we also analyzed mating and oviposition behaviors in *Crz* heterozygous (*+/ΔCrz¹*) females. Notably, neither the remating rate nor egg production in heterozygous females differed significantly from those of wild-type controls (*Figure 1J and K*). These results indicate that the *Crz* loss-of-function phenotype is recessive, and that a single functional copy of *Crz* is sufficient to sustain a normal PMR. Collectively, these results demonstrate that diminished CRZ signaling severely perturbs the PMR in female BPHs,

seen as increased post-mating receptivity and reduced egg-laying. Importantly, the *Crz* deletion abolished the onset of typical post-mating rejection behavior. We, thus, conclude that endogenous CRZ signaling is important for the induction of a PMR in female BPHs.

## CRZ is produced by similar neurons in both sexes of BPHs and is absent from the male accessory gland

To further understand how CRZ regulates the PMR in BPHs, we examined the expression of CRZ and *Crz* across different tissues in males and females. The *Crz* mRNA expression was examined in various BPH tissues using both RT-PCR and qPCR (*Figure 2A*). We detected *Crz* mRNA expression primarily in the central nervous system (CNS), regardless of the primer set or detection method used, while expression was nearly undetectable in other tissues, including reproductive organs of both sexes (*Figure 2B and C*).

A previous study reported CRZ in four male-specific interneurons within the abdominal ganglia of *D. melanogaster* (*Tayler et al., 2012*). To determine whether CRZ exhibits sexually dimorphic expression in BPHs, we performed immunolabeling using an anti-CRZ antiserum. This labeled four pairs of neurons in the protocerebrum of the brain (*Figure 2D and E*) and one pair of neurons in the subesophageal ganglion in both males and females (*Figure 2D and E*). Since CRZ immunolabeled neurons in brains of most, if not all, insects studied include three or more pairs of lateral neurosecretory cells (*Roller et al., 2003*; *Nässel and Zandawala, 2019*), we suggest that CRZ signaling regulating the PMR may be hormonal (at least in part). Importantly, no CRZ-immunopositive neurons were detected in the abdominal ganglia of either sex (*Figure 2D and E*), and thus, there is no evidence for sex-specific expression of CRZ in neurons in BPHs. This is in contrast to *D. melanogaster*, where we confirmed, as previously reported (*Tayler et al., 2012*), the presence of CRZ-immunolabeled neurons specifically in the abdominal ganglia of males (*Figure 2—figure supplement 1A*).

To exclude that CRZ acts as a seminal fluid peptide transferred to females during copulation, similar to sex peptide (SP) in *D. melanogaster* and macc in BPHs (*Zhang et al., 2025*), we specifically examined CRZ expression in reproductive tissues of both sexes. However, we observed no CRZ immunolabeling in the male reproductive organs (*Figure 2F1–F3*) or female reproductive organs (*Figure 2G1–G3*). This aligns with our transcriptional *Crz* data shown above (*Figure 2B and C*). Furthermore, *Crz* mRNA and CRZ protein were absent from transcriptomic and proteomic datasets of the male accessory glands (MAGs) in BPHs (*Zhang et al., 2025*). Collectively, the available data suggest that CRZ is not produced by the MAG and is consequently unlikely to be transferred to females as a seminal fluid component.

## The CRZ receptor is essential for mediating the post mating response in female BPHs

Given our findings implicating CRZ in modulating the PMR in female BPHs, we identified the CRZ receptor (*CrzR*) gene in this species; it encodes a structurally conserved GPCR orthologous to other insect *CrzR*s (*Figure 3—figure supplement 1*). We next explored the effect of knocking down the *CrzR* on the PMR in BPHs. RNAi-mediated knockdown of the receptor (*dsCrzR* injection) resulted in 15% of the mated females re-mating, a phenotype not seen in *dsgfp*-injected controls (*Figure 3A and B*) and in significantly reduced oviposition (*Figure 3C*). Importantly, CRZ injection failed to diminish receptivity in *CrzR*-knockdown virgins (*Figure 3D and E*). Furthermore, MAG extract (with seminal fluid proteins) injection into *dsCrzR*-treated virgins did not affect receptivity (*Figure 3F*).

To further confirm the role of the *CrzR* in the PMR of BPHs, we applied CRISPR/Cas9 genome editing to mutate the *CrzR* gene and constructed a homozygous mutant strain of *CrzR* (*CrzR^M*) by genetic crosses (*Figure 3G–G"*, *Figure 3—figure supplement 2A*). We found that the developmental time and life span of the female *CrzR* mutants, like the *Crz* mutants, were slightly extended compared to wild-type animals (*Figure 3—figure supplement 2E and F*). Homozygous *CrzR^M* females displayed increased remating rates (30%) (*Figure 3I and J*) and the number of eggs laid by *CrzR^M* females was also significantly decreased (*Figure 3K*). We also examined mating and oviposition behaviors in *CrzR* heterozygous (*+/CrzR^M*) females. Notably, both the remating rate and egg production in heterozygous females were comparable to those observed in homozygous *CrzR* mutant (*CrzR^M*) females (approximately 30%) (*Figure 3I and J*). Thus, heterozygosity for the *CrzR* was sufficient to disrupt post-mating behaviors in female BPHs. These findings indicate that the *CrzR* loss-of-function phenotype is

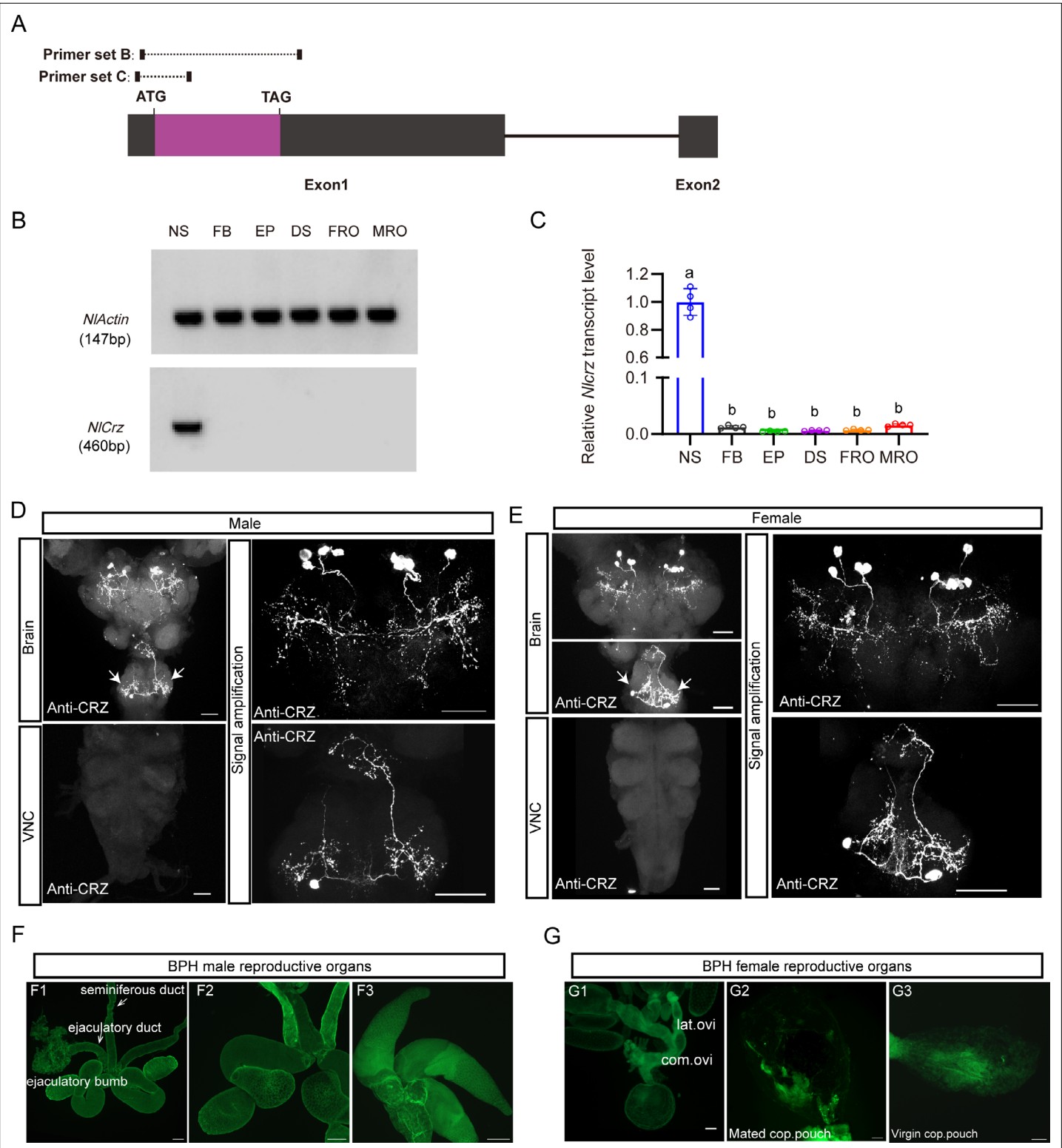

**Figure 2.** Expression of *Crz* and corazonin (CRZ) in the nervous system of the brown planthopper (BPH). (**A**) Schematic of primer design for *Crz* detection (primer sequences provided in ***Supplementary file 1***). (**B**) RT-PCR analysis of *Crz* mRNA levels in various BPH tissues. Actin served as the loading control. Tissues: Nervous system (NS), fat body (FB), Epidermis (EP), digestive system (DS), female reproductive organs (FRO), male reproductive organs (MRO). Note: Tissues except reproductive organs were pooled from both sexes. (**C**) Relative *Crz* expression in female *N. lugens* tissues determined by RT-qPCR (abbreviations as in **B**). Expression was normalized to *Actin* and *18SrRNA* and expressed relative to the mean value of the highest-expressing tissue (set to 1). Data are shown as mean ± s.e.m. Groups that share at least one letter are statistically indistinguishable. One-way

*Figure 2 continued on next page*

*Figure 2 continued*

ANOVA followed by Tukey's multiple comparisons test. (**D-E**) Immunolocalization of CRZ peptide in the nervous system of male (**D**) and female (**E**) BPHs. Four pairs of CRZ neurons in the brain and one pair in the subesophageal ganglion are shown. Note that some of the CRZ neurons are likely to be neurosecretory cells with axon terminations in the corpora cardiaca, others may be interneurons. The extensive branches within the brain suggest CRZ signaling in brain circuits. VNC, ventral nerve cord. Scale bars: 50 μm. (**E**) CRZ immunolabeling in the nervous system of female BPHs. Scale bar: 50 μm. (**F-G**) Lack of CRZ immunoreactivity in BPH reproductive organs. (**F1–F3**) Male reproductive organs (MRO): testes (test). (**G1–G3**) Female reproductive organs (FRO): lateral oviduct (Lat. ov), common oviduct (Com. ov). Scale bars: 100 μm.

The online version of this article includes the following source data and figure supplement(s) for figure 2:

**Source data 1.** Original gel image used for *Figure 2B*.

**Source data 2.** Uncropped gel image for *Figure 2B*, with the relevant bands indicated.

**Source data 3.** Source data for all graphs presented in *Figure 2*.

**Figure supplement 1.** Corazonin (CRZ) immunolabeling in abdominal ganglion and reproductive organs of male and female brown planthoppers (BPHs) and *D. melangaster*.

dominant, and that a single functional copy of *CrzR* is insufficient to maintain a normal post-mating response. Consistent with the *CrzR* RNAi results, neither CRZ nor MAG extract injections reduced receptivity in *CrzR^M* virgins (*Figure 3K–M*). Finally, macc peptide injection also failed to decrease receptivity in *CrzR*-mutant virgins (70% acceptance rate), whereas it significantly suppressed mating in wild-type controls (*Figure 3N*). These results collectively demonstrate that intact CRZ/*CrzR* signaling in female BPHs is required for transducing the PMR-inducing effects of male-derived seminal factors, including macc (*Zhang et al., 2025*).

## The *CrzR* is highly expressed in the reproductive organs of female BPHs

To further elucidate the role of the CRZ signaling in regulation of the PMR, we analyzed the *CrzR* expression in *N. lugens* tissues. We first monitored *CrzR* expression by means of RT-PCR and qPCR. Both methods revealed predominant *CrzR* transcript expression in the female reproductive tract (*Figure 4A–C*). qPCR further demonstrated significantly higher *CrzR* expression in adult females than in males (*Figure 4—figure supplement 1A*), particularly localized to female abdomens (*Figure 4—figure supplement 1B*).

As we failed to generate a functional NlCrzR antibody, to determine the tissue expression of the receptor in more detail, we used in situ hybridization and confirmed strong *CrzR* expression within the female reproductive system (*Figure 4D*). No hybridization signal was detected in the lateral or common oviducts (*Figure 4D1*). However, specific labeling occurred in the sperm storage organs: the spermathecae and pouched glands of the lower reproductive tract (*Figure 4D2 and D2'*). These structures are the primary sites for long-term sperm storage. Specificity was validated by absence of signal in sense-probe controls (*Figure 4D3–D4'*). This enrichment in sperm storage organs suggests that CRZ signaling may modulate post-mating physiology in part by regulating the mobilization of stored sperm and seminal fluid.

Notably, while CrzR is highly expressed in the female reproductive tract, we acknowledge that RT-PCR and qPCR analyses also detected low but detectable CrzR expression in other tissues, including the CNS and fat body (*Figure 4B–C*). Thus, we cannot exclude the possibility that *CrzR* in the brain (e.g. by modulating neural circuits governing reproductive behavior) or fat body (e.g. by affecting post-mating metabolism and nutrient/energy allocation) may also contribute to the PMR. These non-reproductive tract sites of CrzR expression warrant further investigation to fully delineate the systemic roles of CRZ/CrzR signaling in female PMR regulation.

## Endogenous CRZ signaling modulates aspects of the PMR in female *Drosophila melanogaster*

Having established the critical role of CRZ signaling in regulating the PMR of the female BPH, we next investigated its potential function in female reproduction, including the PMR, of the genetically tractable fly *D. melanogaster*.

We first assessed re-mating frequency in *Crz* mutant female flies (*Crz^attp*). Approximately 40% of the *Crz^attp* females that had experienced a first successful copulation mated again, compared to only 3%

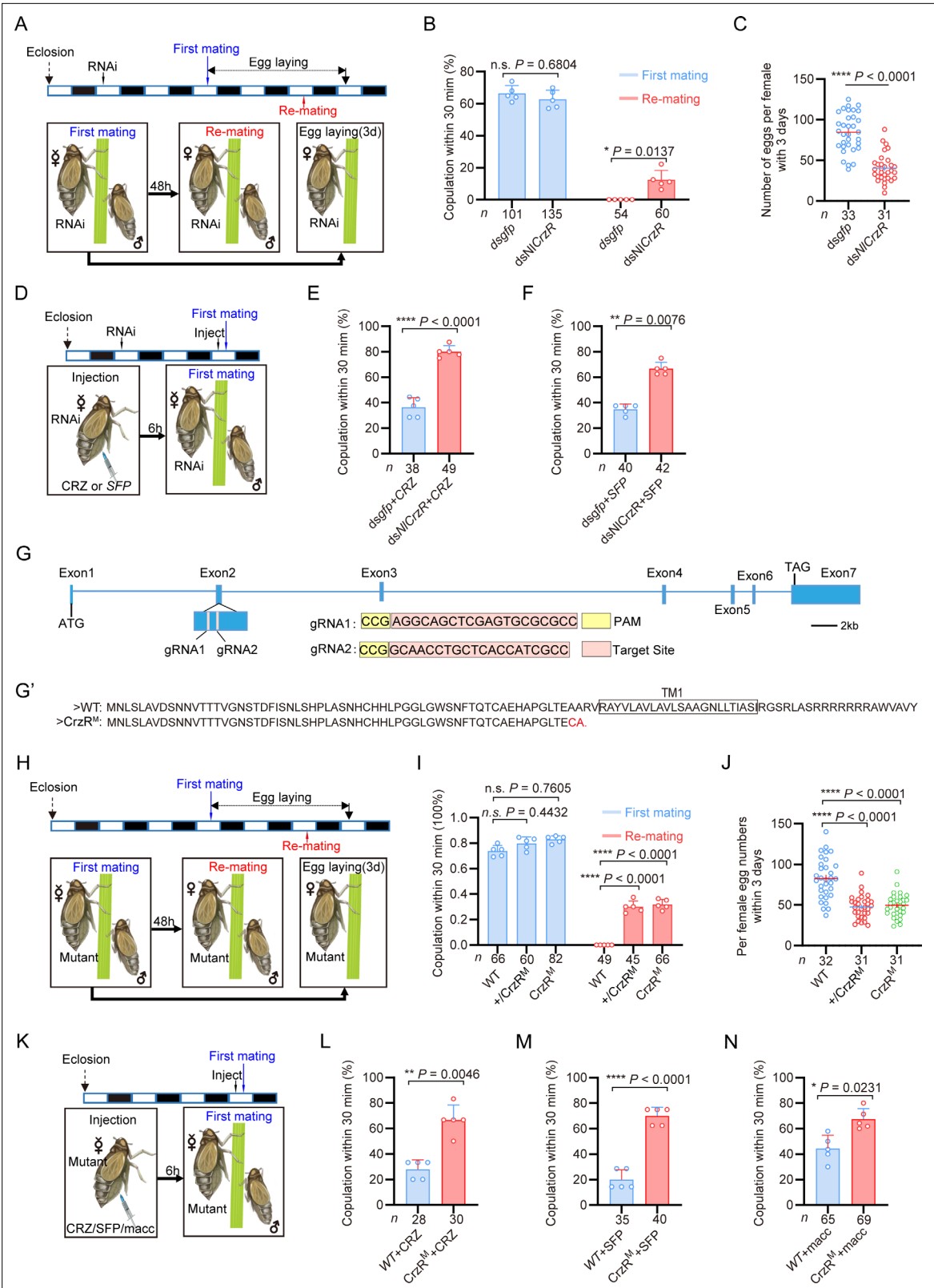

**Figure 3.** The corazonin (CRZ) receptor (*CrzR*) is essential for the female post-mating response (PMR) and is required for action of male accessory gland (MAG)-derived factors. (**A**) Experimental design for **B** and **C**. (**B**) Receptivity after *CrzR*-RNAi (injection of ds*CrzR*; ds*GFP* as control). Percentage of females copulating within 30 min. The small circles denote the number of replicates, ≥9 insects/replicate; the numbers below the bars denote total number of animals. Statistical comparisons of proportions were performed using chi-square tests for contingency tables, *p*<0.05 was considered

*Figure 3 continued on next page*

*Figure 3 continued*

significant. (**C**) Numbers of eggs laid within 72 hr post-mating (*dsCrzR* females × WT males). Data are shown as mean ± s.e.m. \*\*\*\**p*<0.0001 (Student's *t*-test). The numbers below the bars denote total number of animals, ≥4 biological replicates, ≥7 insects/replicate. (**D**) Experimental design for **E-F**. (**E**) Receptivity after CRZ injection in *CrzR*-knockdown virgins (dsGFP+CRZ control). Graph shows the percentage of females copulating within 30 min. Each virgin female BPH was injected with a calibrated glass capillary needle directly into the abdomen with 10 ng of peptide dissolved in 50 nL of 1x PBS balanced salt solution. The small circles denote the number of replicates, ≥7 insects/replicate; the numbers below the bars denote total number of animals. Statistical comparisons of proportions were performed using chi-square tests for contingency tables, *p*<0.05 was considered significant. (**F**) Receptivity after MAG extract (with seminal fluid proteins) injection in *CrzR*-knockdown virgins (dsGFP+SFP control). Percentage copulating within 30 min. The small circles denote the number of replicates, ≥7 insects/replicate; the numbers below the bars denote total number of animals. Statistical comparisons of proportions were performed using chi-square tests for contingency tables, *p*<0.05 was considered significant. (**G**) *CrzR* targeting strategy for generation of mutants. *Top:* Genomic structure (ATG/TAG in Exons 2/7). sgRNA target (pink), PAM (yellow). *Bottom:* CRISPR alleles. (**G′**) Translated receptor mutant sequences. (**H**) Experimental design for **I-J**. (**I**) Receptivity of *CrzR* mutants compared to wild-type animals (WT). Graph displays the percentage of females copulating within 30 min. The small circles denote the number of replicates, ≥9 insects/replicate; the numbers below the bars denote total number of animals. Statistical comparisons of proportions were performed using chi-square tests for contingency tables, *p*<0.05 was considered significant. (**J**) Number of eggs laid by mutants mated with WT males within 72 hr. The numbers below the bars denote total number of animals, ≥4 biological replicates, ≥7 insects/replicate. Data are shown as mean ± s.e.m. \*\*\*\**p*<0.0001; Student's *t*-test. (**K**) Experimental design for **L-M**. (**L**) Receptivity after CRZ injection in *CrzR* mutants compared to WT. Graph displays the percentage of females copulating within 30 min. Each virgin female BPH was injected with a calibrated glass capillary needle directly into the abdomen with 10 ng of peptide dissolved in 50 nL of 1x PBS balanced salt solution. The small circles denote the number of replicates, ≥5 insects/replicate; the numbers below the bars denote total number of animals. Statistical comparisons of proportions were performed using chi-square tests for contingency tables, *p*<0.05 was considered significant. (**M**) Receptivity after SFP injection in *CrzR* mutants compared to WT. Graph displays the percentage of females copulating within 30 min. The small circles denote the number of replicates, ≥7 insects/replicate; the numbers below the bars denote total number of animals. Statistical comparisons of proportions were performed using chi-square tests for contingency tables, *p*<0.05 was considered significant. (**N**) Receptivity after macc injection in *CrzR* mutants compared to WT. Graph displays the percentage of females copulating within 30 min. The small circles denote the number of replicates, ≥10 insects/replicate; the numbers below the bars denote total number of animals. Statistical comparisons of proportions were performed using chi-square tests for contingency tables, *p*<0.05 was considered significant.

The online version of this article includes the following source data and figure supplement(s) for figure 3:

**Source data 1.** Source data for all graphs presented in *Figure 3*.

**Figure supplement 1.** Structural conservation of the *CrzR* across insect species.

**Figure supplement 2.** Characterization of *CrzR* mutants in *N. lugens*.

**Figure supplement 2—source data 1.** Source data for all graphs presented in *Figure 3—figure supplement 2*.

of mated wild-type control females (*Figure 5A and B*). Importantly, the *Crz* mutation did not affect the first mating of females (*Figure 5B*). Furthermore, mated *Crz^attp^* females laid approximately 45% fewer eggs than mated wild-type females (*Figure 5C*). These results demonstrate that CRZ signaling is necessary for aspects of the PMR in female *D. melanogaster*, specifically inhibiting re-mating and stimulating post-mating egg production. Immunohistochemical labeling confirmed the absence of CRZ protein in *Crz^attp^* mutants (*Figure 5D*).

To determine whether release of CRZ from *Crz* neurons is responsible for inhibition of remating and stimulation of oviposition, we knocked down *Crz* expression specifically in CRZ-producing neurons using the GAL4-UAS system (*Crz-GAL4>UAS-Crz-RNAi*). Consistent with the mutant phenotype, 22% of the mated *Crz-GAL4>UAS-Crz-RNAi* females mated again after their first successful mating with wild-type males, whereas only 2% of the control females (*Crz-GAL4/+*) did so (*Figure 5E*). Virgin receptivity was again unaffected by *Crz* knockdown (*Figure 5E*). Additionally, mated *Crz-GAL4>UAS-Crz-RNAi* females laid approximately 20% fewer eggs than mated controls (*Crz-GAL4/+*) (*Figure 5F*). To further confirm the role of CRZ neuropeptide in modulation of the PMR in females, we injected this peptide in virgin females. Consistent with the above results, CRZ-injected virgins displayed a reduced response to mating attempts (*Figure 5G*). To determine whether CRZ modulates the PMR endogenously also in female *D. melanogaster*, we first examined whether the male accessory gland (MAG) expresses *Crz*. RT-PCR analysis revealed no detectable *Crz* transcripts in the *D. melanogaster* MAG (*Figure 5H*), indicating that the MAG does not synthesize the CRZ peptide. To further exclude the possibility that CRZ or any other *Crz*-precursor-derived peptide might enter the AG from other sources, we used the GAL4–UAS system to visualize *Crz*/CRZ-producing neurons and performed immunolabeling with anti-CRZ antibodies. Neither approach revealed any signal in the MAG (*Figure 5I*). Some GFP signals were observed in cells of the testes (indicated by white arrows), whereas no corresponding signal was detected by CRZ immunohistochemistry (*Figure 5I*).

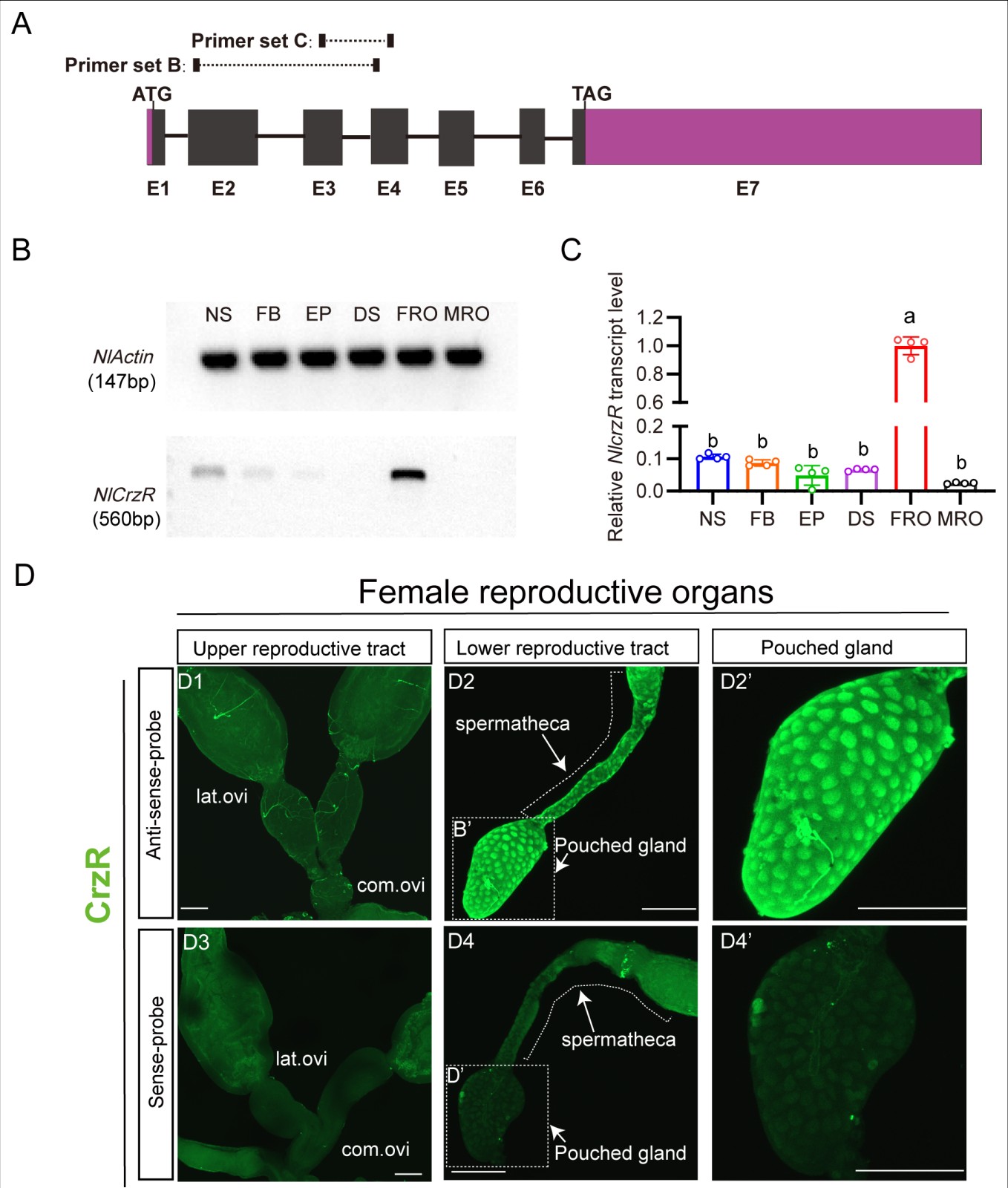

**Figure 4.** Spatial expression profiling of *CrzR* in the female *N. lugens* reproductive tract. (**A**) *CrzR* genomic structure with primer design for experiments in panels **B**-**C**. (**B**) RT-PCR analysis of *CrzR* tissue distribution. Note weaker expression in central nervous system and fat body. Actin loading control is shown in top row. Tissue abbreviations: NS: central nervous system; FB: fat body; EP: Epidermis; DS: digestive system, FRO: female reproductive organs, MRO: male reproductive organs. Except for the reproductive system, the rest of the tissues come from a mixture of female and male. (**C**) Relative *CrzR*

*Figure 4 continued on next page*

*Figure 4 continued*

expression in female *N. lugens* tissues determined by RT-qPCR (abbreviations as in **B**). Expression was normalized to *Actin* and *18SrRNA* and expressed relative to the mean value of the highest-expressing tissue (set to 1). Data are shown as mean ± s.e.m. Groups that share at least one letter are statistically indistinguishable. One-way ANOVA followed by Tukey's multiple comparisons test. (**D1**-**D2'**) *CrzR* mRNA localization via fluorescent in situ hybridization (antisense probe): D1: There is no signal in lateral/common oviducts; D2/D2': Specific signal is detected in spermatheca (SP) and pouched gland (PG). (**D3**-**D4'**) Negative controls (sense probe) showing background-level signal. Scale bars: 100 μm.

The online version of this article includes the following source data and figure supplement(s) for figure 4:

**Source data 1.** Original gel image used for *Figure 4B*.

**Source data 2.** Uncropped gel image for *Figure 4B*, with the relevant bands indicated.

**Source data 3.** Source data for all graphs presented in *Figure 4*.

**Figure supplement 1.** Developmental and tissue-specific expression profiling of *CrzR* in *N. lugens* determined by qPCR.

**Figure supplement 1—source data 1.** Source data for all graphs presented in *Figure 4—figure supplement 1*.

To further evaluate whether the MAG could be a source of CRZ, we interrogated publicly available single-cell transcriptomic datasets from the Fly Cell Atlas covering the male reproductive glands (including MAG main and secondary cells, ejaculatory duct/bulb epithelium, secretory and muscle cells) and the male testis (*Li et al., 2022*; *Figure 5—figure supplement 1*). In the MAG atlas, *Crz* transcripts were undetectable across MAG main and secondary cells as well as other gland-associated cell types (*Figure 5—figure supplement 1A*). In contrast, the MAG-derived sex peptide (SP) was robustly detected, confirming correct tissue representation and cell-type annotation (*Figure 5—figure supplement 1A*). In the testis atlas, Crz expression was sparse and restricted to a small subset of cells rather than enriched in any major spermatogenic lineage (*Figure 5—figure supplement 1B and C*), whereas β2-tubulin prominently marked spermatogenic populations (*Figure 5—figure supplement 1B*). Hence, these single-cell data support our RT-PCR and anti-CRZ immunolabeling findings. Together, our data demonstrate that the male accessory gland does not produce or sequester CRZ. In conclusion, like in BPHs, CRZ acts endogenously in females to modulate post-mating behavior also in *D. melanogaster*.

## Discussion

Our study unveils a novel function of CRZ in an endogenous female signaling pathway that modulates the post-mating reproductive behavior and physiology (collectively PMR) in BPHs. It is, thus, the first report linking CRZ to a PMR in insects. In contrast to SP and DUP99B in *D. melanogaster* (*Kubli, 2003*; *Chen et al., 1988*), HP-1 in *Aedes aegypti* (*Duvall et al., 2017*) and maccessin (macc) recently identified in BPHs (*Zhang et al., 2025*), CRZ is not produced in the MAG, and thus not transferred from males during copulation. Instead, it is acting as an endogenous peptide derived from neuroendocrine cells in the female brain. Moreover, this CRZ-mediated endogenous modulation in females is also seen in *D. melanogaster*. When *Crz* is knocked down or knocked out in flies, mated females likewise exhibit an altered PMR.

Focusing on the endogenous CRZ signaling in female BPHs, our main findings using mutants and RNAi of both *Crz* and *CrzR*, demonstrate that CRZ signaling decreases the female responsiveness to courting males, especially in already mated females. Also, oviposition in mated BPHs is increased by CRZ signaling. CRZ immunolabeling revealed a small set of neurons in the brain and subesophageal ganglion and no CRZ was detected in reproductive organs. Thus, the likely source of CRZ affecting the PMR is the CRZ-producing neuroendocrine cells in the brain. In insects studied so far (including other hemipterans), there are at least three pairs of CRZ expressing lateral neurosecretory cells with axon terminations in the corpora cardiaca (*Roller et al., 2003*; *Nässel and Zandawala, 2019*), suggesting that in BPHs and *D. melanogaster* hormonal CRZ may contribute to the modulation of the PMR. However, local CRZ signaling in brain circuits cannot be excluded since the CRZ neurons arborize extensively within the brain.

We found that the CRZ receptor, *CrzR*, is predominantly expressed in the female reproductive tract in BPHs, more specifically in the lower tract, including the spermatheca and pouched gland, which are sites of sperm and seminal fluid storage. Thus, in BPHs, CRZ may act to regulate the release of stored sperm and seminal fluid in the female. This regulation would also gate the flow of male-derived

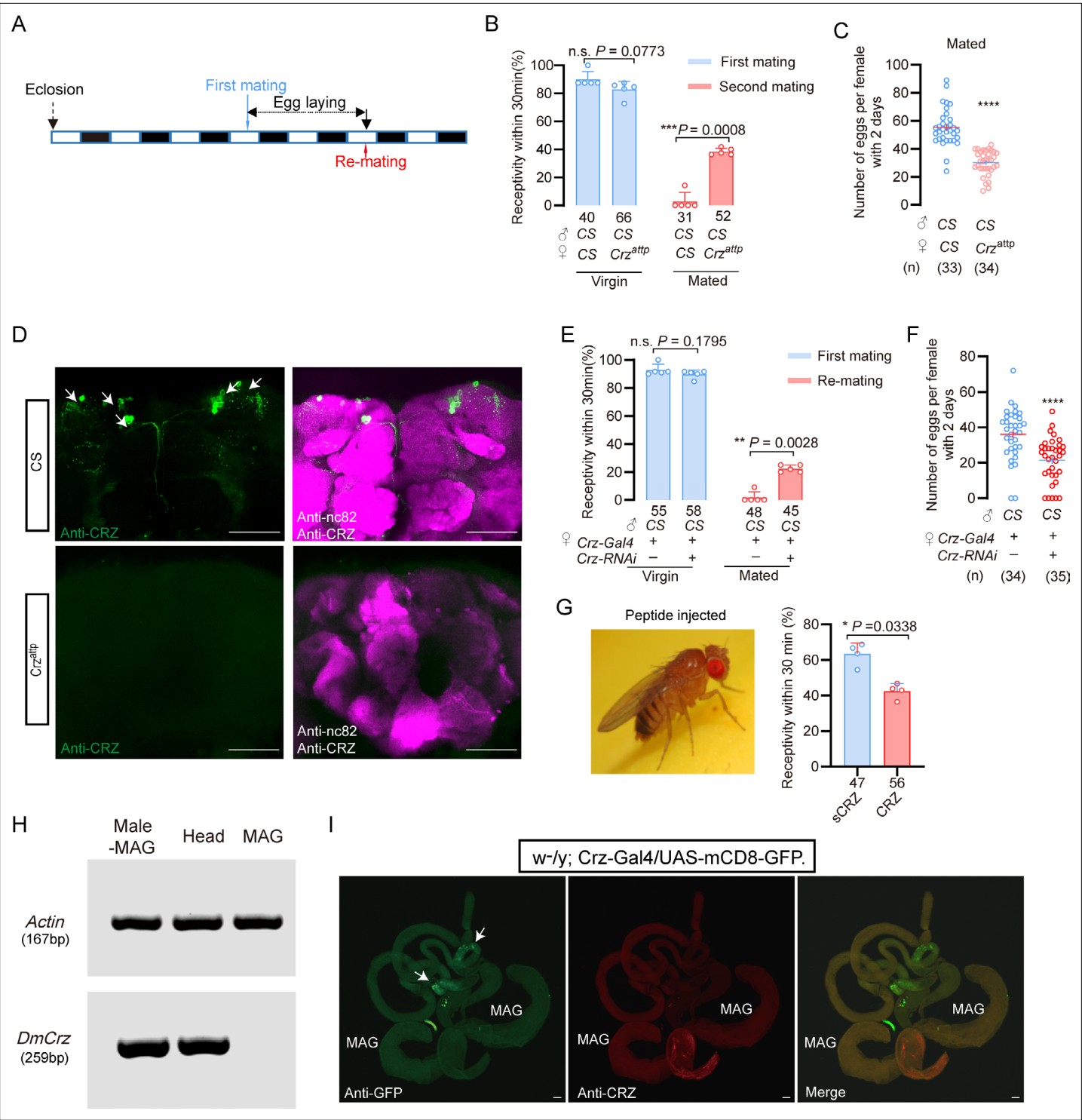

**Figure 5.** Corazonin (CRZ) regulates part of the post-mating response (PMR) in *D. melanogaster*. (**A**) Schematic of experimental design. White segments on the time axis indicate light periods; black segments indicate dark periods. (**B**) Receptivity of virgin and mated *Crz* mutant (*Crz*^attp^) females. Graph displays the percentage of females copulating within 30 min. Small circles indicate the number of replicates, ≥6 insects/replicate; numbers below bars denote total number of animals. Statistical comparisons of proportions were performed using chi-square tests for contingency tables, *p*<0.05 was considered significant. (**C**) Graph shows eggs laid per female after mating. Numbers below bars denote total animals, ≥4 biological replicates, ≥8 insects/replicate. ****p<0.0001 (Student's t-test). (**D**) Anti-CRZ immunostaining in brains of *Crz* mutant (*Crz*^attp^), control (Cs), and after *Crz*-RNAi. Note CRZ-expressing cells at arrows in control. Scale bar: 100 μm. (**E**) Receptivity of virgin and mated females after *Crz*-RNAi. Graph displays the percentage of females copulating within 30 min. Small circles indicate the number of replicates, ≥8 insects/replicate; numbers below bars denote total animals.

*Figure 5 continued on next page*

*Figure 5 continued*

Statistical comparisons of proportions were performed using chi-square tests for contingency tables, *p*<0.05 was considered significant. (**F**) Graph shows eggs laid per female after mating. Numbers below bars denote total animals, ≥4 biological replicates, ≥8 insects/replicate. ****$p$<0.0001 (Student's *t*-test). (**G**) Receptivity of virgin females following CRZ peptide injection (monitored 6 h after injection). Graph displays the percentage of females copulating within 30 min. Small circles indicate the number of replicates, ≥9 insects/replicate; numbers below bars denote total animals. Statistical comparisons of proportions were performed using chi-square tests for contingency tables, *p*<0.05 was considered significant. (**H**) RT-PCR analysis of Crz tissue distribution. Note the lack of expression in the accessory gland. Actin loading control is shown in the top row. Tissue abbreviations: MAG, male accessory gland; Male–MAG, whole body without male accessory gland. (**I**) Anti-GFP and anti-CRZ immunostaining in brains of *w⁻/y; Crz-Gal4/UAS-mCD8-GFP*. Note no signal was detected in male accessory gland (MAG). Some GFP-stained signals were observed in the testes (indicated by white arrows), whereas no corresponding signals were detected via CRZ antibody staining. Scale bar: 50 µm.

The online version of this article includes the following source data and figure supplement(s) for figure 5:

**Source data 1.** Original gel image used for *Figure 5H*.

**Source data 2.** Uncropped gel image for *Figure 5H*, with the relevant bands indicated.

**Source data 3.** Source data for all graphs presented in *Figure 5*.

**Figure supplement 1.** Crz expression is undetectable in the *Drosophila* male accessory gland in Fly Cell Atlas single-cell datasets.

factors that induce and maintain the PMR, such as macc and ITPL-1. As mentioned, we cannot exclude additional action of CRZ in neuronal circuits in the CNS that regulate female sexual arousal and reproductive behavior, since our PCR data indicate expression of the *CrzR* also in the CNS. Additionally, the fat body of BPHs, express *CrzR*, similar to *D. melanogaster* (*Kubrak et al., 2016*; *Li et al., 2022*), possibly suggesting a role of CRZ in post-mating metabolism.

The brain CRZ neurons of BPHs are likely to be activated by sensory signals from the female reproductive tract upon copulation and hence release CRZ into the circulation (and maybe within the brain). In *D. melanogaster,* sensory neurons in the reproductive tract are activated by SP (via the SP receptor), and signal via a chain of ascending axons of the VNC to sex-specific neurons in the brain to induce the PMR switch in behavior and physiology (*Yang et al., 2009*; *Häsemeyer et al., 2009*; *Auer and Benton, 2016*; *Nallasivan et al., 2024*). Thus, in *D. melanogaster,* these sex-specific circuits may also activate the CRZ neurons. Possibly similar sensory and sex-specific neurons exist in BPHs and respond either to mechanical stimuli or factors in the seminal fluid leading to activation of CRZ neurons and other brain neurons that regulate the PMR.

The female PMR is complex in *D. melanogaster* and not only receptivity and oviposition are affected, but also feeding, metabolism, sleep patterns, and aggression (*Kubli, 2003*; *Wolfner, 2002*; *Chen et al., 1988*; *Ellendersen and von Philipsborn, 2017*; *Isaac et al., 2010*; *Bath et al., 2017*). In BPHs, only the receptivity (and associated rejection behavior) and oviposition have been analyzed in any detail. We found that the MAG-derived peptides transferred with the semen, macc and ITPL-1, only affected the receptivity (*Zhang et al., 2025*), whereas we show here that CRZ also affects oviposition. Thus, we have identified three different peptides CRZ, macc, and ITPL-1 that all influence the PMR in different ways, suggesting complex regulatory mechanisms also in BPHs.

It is interesting to note that CRZ is an evolutionarily conserved neuropeptide that has been found in most, but not all, insect species and it displays a wide array of functions (*Nässel and Zandawala, 2019*; *Kubrak et al., 2016*; *Predel et al., 2007*; *Boerjan et al., 2010*). The CRZ function varies across different insect species, but a common role seems to be in regulation of stress (*Predel et al., 2007*; *Boerjan et al., 2010*; *Veenstra, 2009*; *Zhao et al., 2010*). Other functions are for example: in the American cockroach, CRZ accelerates the heart rate (*Veenstra, 1989*; *Zhao et al., 2010*), in locusts, it triggers the formation of pigments that darken the body color in gregarious insects (*Tawfik et al., 1999*; *Veenstra, 1989*), in the moth *Bombyx mori*, CRZ affects developmental speed and reduces the rate of silk spinning behavior (*Tanaka et al., 2002*; *Tawfik et al., 1999*); in *D. melanogaster* larvae, CRZ regulates molting motor behavior (*Kim et al., 2004*; *Tanaka et al., 2002*) and growth (*Imura et al., 2020*; *Kim et al., 2004*); in adult *D. melanogaster* CRZ regulates stress responses, food intake and metabolism (*Kubrak et al., 2016*; *Zandawala et al., 2021*), and protein food preference (*Liu et al., 2024*), as well as sperm ejection and mating duration in males (*Tayler et al., 2012*); and in social ants, it regulates the social hierarchy and inhibits egg-laying in the queen (*Gospocic et al., 2017*). Finally, a recent study found that CRZ mediates a diapause-induced suppression of reproduction in female bean bugs (*Zandawala et al., 2021*; *Xi et al., 2025*). To all these functions can now be added the role of CRZ in reproductive behavior and physiology of female BPHs and *D. melanogaster*. Finally, building

on the role of CRZ in metabolism shown in *D. melanogaster* (*Li et al., 2022*; *Kubrak et al., 2016*), it is possible that CRZ also regulates post-mating metabolism in female BPHs and flies. Thus, there may be a link between CRZ and the altered feeding and metabolism commonly seen in mated female insects that ensures optimal egg development (*Kubli, 2003*; *Camus et al., 2018*; *Wolfner, 2002*).

CRZ and the vertebrate peptide gonadotropin-releasing hormone (GnRH) are paralogs that arose via an ancient gene duplication, and act on evolutionarily conserved receptors (*Tian et al., 2016*; *Yañez Guerra and Zandawala, 2023*). *Bona fide* GnRHs have not been found in arthropods and the CRZ peptide has been lost in the vertebrate lineage, but the cephalochordate (lancelet) *Branchiostoma floridae* does have peptides of both types (*Yañez Guerra and Zandawala, 2023*; *Tian et al., 2016*). In vertebrates, including humans, GnRH regulates female reproduction (specifically sexual maturation) (*Knobil, 1988*; *Millar, 2005*). We have shown here that CRZ regulates reproductive behavior and physiology in females of *D. melanogaster* and previous reports demonstrated a role in female fecundity (*Bergland et al., 2012*; *Millar, 2005*) and a reproductive function also in males of this species (*Tayler et al., 2012*). Thus, CRZ and GnRH share a major function that may be ancient, although CRZ signaling in insects has evolved to also regulate stress responses, metabolism, and development (*Kubrak et al., 2016*; *Boerjan et al., 2010*; *Barredo et al., 2020*).

Some outstanding questions remain with respect to mechanisms of CRZ signaling in the female PMR. The inputs triggering release of hormonal CRZ at mating are not known in BPHs or flies. This can probably be best investigated in *Drosophila* where brain neurons that are targets of the sex peptide receptor-expressing neurons in the oviduct have been identified (*Nallasivan et al., 2024*; *Kubrak et al., 2016*), and synaptic inputs to CRZ-producing brain neurons have been described (*McKim et al., 2024*; *Barredo et al., 2020*). Once candidate interneurons interacting with the CRZ neurons have been identified, genetic manipulations of these neurons may unveil how mating triggers endogenous CRZ signaling and the PMR. Another aspect of CRZ signaling that merits further study in *Drosophila* is the link between CRZ and the post-mating change in feeding and metabolism. At present, it is not known whether also BPHs change their feeding and metabolism after mating, but it would be valuable to investigate to determine to what extent the PMR in the two insects are similar. Finally, since BPHs are a major rice pest, it may be useful to further investigate the regulation of the PMR to design novel strategies for control their reproduction.

## Key resources table

| Reagent type (species) or resource | Designation | Source or reference | Identifiers | Additional information |
| --- | --- | --- | --- | --- |
| Gene (*Nilaparvata lugens*) | crz | GenBank | GenBank accession: AB817247.1 | |
| Gene (*N. lugens*) | crzr | GenBank | GenBank accession: AB817313.1 | |
| Strain, strain background (*N. lugens*) | Wild-type brown planthopper | Lab | Collected from Hangzhou, Zhejiang Province, China, in 1995; maintained on Taichung Native 1 (TN1) rice seedlings | |
| Genetic reagent (*Drosophila melanogaster*) | Canton-S | Bloomington *Drosophila* Stock Center | BDSC:64349 | Wild-type strain |
| Genetic reagent (*D. melanogaster*) | CrzattP | Bloomington *Drosophila* Stock Center | BDSC:84487 | FBst0084487 |
| Genetic reagent (*D. melanogaster*) | Crz-GAL4 | Bloomington *Drosophila* Stock Center | BDSC:51976 | FBst0051976 |
| Genetic reagent (*D. melanogaster*) | Crz-RNAi | Tsinghua Fly Center | THU:2139 | RNAi line targeting Crz; |

*Continued on next page*

*Continued*

| Reagent type (species) or resource | Designation | Source or reference | Identifiers | Additional information |
|---|---|---|---|---|
| Antibody | anti-CRZ (Rabbit polyclonal) | Gift from Prof. Jan A. Veenstra | | IF(1:1000); used for immunostaining |
| Antibody | anti-Bruchpilot / nc82 (Mouse monoclonal) | Developmental Studies Hybridoma Bank | Cat# nc82; RRID:AB_2314866 | IF(1:1000) |
| Antibody | donkey anti-rabbit IgG conjugated to Alexa 488 | Thermo Fisher Scientific | Cat#R37118 | IF(1:500) |
| Antibody | donkey anti-mouse IgG conjugated to Alexa 555 | Thermo Fisher Scientific | Cat#R37115 | IF(1:500) |
| Antibody | anti-DIG conjugated fluorescent antibody | Jackson | Cat#200-542-156 | IF(1:100) |
| Peptide, recombinant protein | Corazonin (CRZ) | GenScript | | Sequence: pQTFQYSRGWNamide |
| Peptide, recombinant protein | Scrambled CRZ peptide | GenScript | | Sequence: pQSRYTTFGQWTNamide |
| Peptide, recombinant protein | TrueCut Cas9 Protein v2 | Thermo Fisher Scientific | Cat#A36497 | Used for CRISPR/Cas9 mutagenesis in N. lugens embryos |
| Sequence-based reagent | dsRNA for NlCrz | GenScript | | F: TAATACGACTCACTATAGGGGCTAGTGTTGCAATGCTGTTG R: TAATACGACTCACTATAGGGGCTGTTGAGCGTTAGACTGT |
| Sequence-based reagent | dsRNA for NlCrzR | GenScript | | F: TAATACGACTCACTATAGGGCGCCGTCTACACACTCATCT R: TAATACGACTCACTATAGGGCTACCAGCTTCGTACAGCGT |
| Sequence-based reagent | sgRNA1 for NlCrz | GenScript | | F: TAATACGACTCACTATAGGCACAGGAGTGATGCAGAC R: TTCTAGCTCTAAAACGTCTGCATCACTCCTGTGCC |
| Sequence-based reagent | sgRNA2 for NlCrz | GenScript | | F: TAATACGACTCACTATAGCTACTAGTGCTGGGCTGCA R: TTCTAGCTCTAAAACTGCAGCCCAGCACTAGTAGC |
| Sequence-based reagent | sgRNA1 for NlCrzR | GenScript | | F: TAATACGACTCACTATAGGCACGCACTCGAGCTGCCT R: TTCTAGCTCTAAAACAGGCAGCTCGAGTGCGTGCC |
| Commercial assay or kit | TRIzol reagent | Invitrogen | CAT#15596026 | Total RNA extraction |
| Commercial assay or kit | M-MLV reverse transcription kit | Biotech | CAT#B639277 | cDNA synthesis for cloning/RT-PCR |
| Commercial assay or kit | HiScript II Q RT SuperMix for qPCR (+gDNA wiper) kit | Vazyme | CAT#R223-01 | First-strand cDNA synthesis |
| Commercial assay or kit | UltraSYBR Mixture (with ROX) Kit | CWBIO | CAT#CW0957M | Real-time qPCR |
| Commercial assay or kit | MEGAscript T7 transcription kit | Ambion | CAT#AM1333 | dsRNA synthesis |

*Continued on next page*

*Continued*

| Reagent type (species) or resource | Designation | Source or reference | Identifiers | Additional information |
|---|---|---|---|---|
| Commercial assay or kit | GeneArt Precision gRNA Synthesis Kit | Thermo Fisher Scientific | CAT#29377 | sgRNA synthesis |
| Commercial assay or kit | Pre-hybridization solution | Boster | Cat#AR0152 | Used for FISH |
| Commercial assay or kit | DIG RNA Labeling Kit (SP6/T7) | Roche | Cat#11175025910 | Synthesis of in situ hybridization probes |
| Software, algorithm | Fiji | ImageJ | RRID:SCR_002285 | Image processing |
| Software, algorithm | GraphPad Prism 9 | GraphPad | RRID:SCR_002798 | Statistical analysis and graphing |
| Software, algorithm | Adobe Illustrator | Adobe | RRID:SCR_010279 | Figure preparation |
| Software, algorithm | ZEN | Zeiss | RRID:SCR_013672 | Confocal image acquisition |
| Software, algorithm | Motic Images Plus 3.0 | Motic | | Microscopy image acquisition/processing |

## Methods

### Experiment insects

All the brown planthoppers used in this research were collected from Hangzhou, Zhejiang Province in 1995. The insects were kept indoors on fresh rice seedlings (Taichung Native 1, TN1) in a growth chamber at 27±1 °C and 70±10% relative humidity, under a 16 hr:8 hr (Light: Dark) photoperiod (*Wu et al., 2017*; *McKim et al., 2024*).

All *D. melanogaster* stocks were raised in standard cornmeal–molasses–agar medium maintained at 25 °C, 60% humidity, and under 12 hr:12 hr light:dark conditions. We used Canton's flies as the wild-type strain. The relevant information about fruit flies used in the experiments are shown in key resources table.

### Male accessory gland protein extraction of *Nilaparvata lugens*

Protein extraction was performed as described previously (*Zhang et al., 2025*). Male accessory glands (MAGs) were dissected from 3 to 5 day-old virgin males in phosphate buffer (pH 7.2) under a stereo-microscope. For protein extraction, approximately 300 MAGs were collected, immediately frozen, and stored at −80 °C until use. Frozen MAGs were pre-ground in liquid nitrogen and then lysed in 500 μL protein cracking buffer (100 mM $NH_4HCO_3$, 8 M urea, 0.2% SDS, pH 8.0) with thorough homogenization. Lysates were centrifuged at 12,000×g for 15 min at 4 °C, and the supernatants were collected as soluble protein extracts. Protein concentration was determined using a Bradford colorimetric assay (BSA as the standard; absorbance at 595 nm). Protein integrity and quality were assessed by SDS-PAGE. Protein extracts were stored at −80 °C until next step experiment.

### Behavior assays of *Nilaparvata lugens*
#### Mating and receptivity

All BPHs were raised in incubators with a 16 hr:8 hr dark:light cycle. Unmated female and male BPHs were collected within 12 hr of emergence. For RNAi or neuropeptide injection treatment, the protocol is shown in *Figure 1A and D*. The mating behavior (or female receptivity behavior) test in this study usually conducted the first mating between 3 and 5 days after emergence. The mating experiment was completed in a plastic tube (2.5 cm in diameter and 10 cm in height). Fresh rice seedlings of appropriate length were placed in each tube, and then a female and a male BPH was added. The

mating conditions and other reproductive behavior parameters during this period were observed in a constant temperature room at 25 °C for 30 min, and the males were taken out after 30 min. In *N. lugens*, mating induces a transient suppression of female receptivity that is not permanent; females typically start to regain remating willingness 72 hr after the first mating, as documented in our previous study (*Zhang et al., 2025*). This temporal window guided the design of our remating assays, in which females were paired with naive males at 48 hr, post-initial mating to capture both the suppressed and recovered phases of receptivity. For the experiment requiring a second mating, females successfully mated for the first time were retained. After 48 hr, wild-type males were put in for the re-mating test. The males in all secondary mating experiments were wild-type. The monitoring time for each mating behavior was 30 min; the receptivity rate refers to the proportion of successful matings of female individuals within the 30 min measurement period.

## Egg laying

After completing the first mating, the male is sucked out and a sufficient number of rice seedlings are added to the tube, and the female is retained to lay eggs in the tube. After 72 hr, the spawning female is transferred out of the tube. After the eggs have matured, about 4–5 days later, we used tweezers to gently manipulate the rice seedlings and counted the number of fertilized eggs produced by the females. Note that 4–5 days after the egg is produced, the successfully fertilized egg will display red eye-spots. Oviposition assay protocol: For quantitative assessment of post-mating responses, egg-laying activity was monitored over a 72 hr observation window following verified initial copulation. To eliminate confounding effects from sequential mating trials, experimental subjects undergoing oviposition measurement were permanently excluded from subsequent re-mating evaluations.

To assess oviposition in females injected with CRZ peptide, virgin or mated female BPHs (3–5 days post-emergence) were transferred into plastic tubes containing sufficient rice seedlings 6 hr after recovery from peptide injection. Females were allowed to oviposit for 24 hr and were then removed. Eggs were subsequently counted by gently dissecting the rice seedlings with forceps.

## Behavior assays of *D. melanogaster*

Mating and re-mating: During the receptivity assessment, all experimental fruit flies were virgin flies that emerged within 12 hr in a 20 °C incubator in a 12 hr:12 hr dark:light cycle. After collection, they were kept at 25 °C incubator in a 12 hr:12 hr dark:light cycle for 4 days before behavioral testing. Receptivity behavior measurement: a perforated plate was placed on top of a non-perforated plate. Male fruit flies were briefly anesthetized on ice and gently transferred into the holes of the perforated plate, with one male per hole. Once the male transfer was completed, a plastic film was gently placed over the perforated plate, followed by covering it with another perforated plate. Then, the virgin female fruit fly, also anesthetized on ice, was placed in each hole. After the transfer was completed, a non-perforated plate was placed on top. This setup was then transferred to a 25 °C incubator to acclimate for 30 min The plastic film separating the male and female flies was quickly removed, and video recording began for 30 min. The recorded videos were analyzed to calculate the acceptance rate of the females. Successful mating females were collected and individually transferred into plastic tubes containing sufficient artificial food for fruit flies to lay eggs. After 24 hr of laying eggs, the female flies were transferred to a new finger-shaped tube with ample artificial feed for another 24 hr of laying. After 48 hr, these egg-laying females were subjected to a secondary mating assessment with wild-type, unmated males, using the same method as described above, to evaluate the acceptance rate during the second mating process. Additionally, the number of eggs laid by each female during the 48 hr was recorded.

Egg laying for mated females: First, collect virgin female fruit flies that eclosion within 12 hr at 20 °C incubator in a 12 hr:12 hr dark:light cycle and then keep them in a 25 °C incubator in a 12 hr:12 hr dark:light cycle for 4 days. After that, unmated males of the wild-type and virgin females were mixed 1:1.5 times. 24 hr after full mating, female flies were lightly anesthetized with ice and transferred to plastic vials containing enough artificial fruit fly food for them to lay eggs for 24 hours. After 24 hr, the flies were transferred again to a new bottle full of artificial food for another 24 hr of egg laying. The number of eggs in each vial is calculated under a microscope, and the total number of eggs in the two vials of each female fruit fly represents the egg-laying capacity of that individual.

## Developmental duration statistics

Nymphs of BPHs that hatched synchronously from distinct genotypes were collected and reared in plastic cups containing fresh rice seedlings; the seedlings were replaced regularly to maintain optimal conditions. From the day the nymphs reached the fifth instar, cups were inspected daily for adult emergence, and the number of newly emerged adults was recorded. After all surviving individuals had completed eclosion, the total nymph to adult developmental duration was calculated.

## BPHs survival rate statistics

For neuropeptide injection: After 4 day emergence, virgin BPHs were injected with CRZ neuropeptide or scrambled CRZ, and after 6 hr of recovery, these insects were placed into a plastic tube with sufficient quantity of rice seedlings. Survival was monitored every 24 hr until all specimens were dead.

For mutant: Wild-type and mutant BPHs adults that eclosed on the same day were individually confined in plastic tubes, each supplied with ample fresh rice seedlings and water renewed regularly to ensure optimal living and nutritional conditions. The number of surviving individuals of each genotype was recorded every 24 hr until all had died, and survival curves were generated from these data.

## Neuropeptide synthesis and injection

Corazonin (CRZ) and scrambled CRZ (sCRZ) were synthesized by Genscript Co., Ltd (Nanjing, China). Peptide masses were confirmed by Mass spectrometry and the amount of peptide was quantified by amino acid analysis. The amino acid sequences of the peptides used in this study are: *N. lugens* corazonin: pQTFQYSRGWNamide; scrambled corazonin: pQSRYTTFGQWTNamide. On the fourth day after adult emergence, virgin female BPHs were injected with different concentrations of synthetic CRZ peptide and control scrambled peptide. The scrambled peptide control was used to monitor for potential non-specific effects on behavior due to the process of injection. Each mg peptide was dissolved in 100 µl DMSO solution (Acmec, #D54264), and then diluted to 1.5 mM, 0.15 mM, 0.015 mM, and 0.0015 mM with 1xPBS as solvent. The BPHs were anesthetized with $CO_2$ for about 30 seconds and placed abdomen up on 2% agarose (Solarbia, #A8201) plate and 50 nl neuropeptide solution injected into each female. Six hours after neuropeptide injection, the insects were used for behavior assay.

## Gene cloning and sequence analysis

We used the NCBI database with BLAST programs to carry out sequence alignment and analysis. Then we predicted Open Reading Frames (ORFs) with EditSeq. The primers were designed by Primer designing tool in NCBI. Total RNA Extraction was using the TRIzol reagent (Invitrogen, Carlsbad, CA, USA) according to the manufacturer's instructions. The cDNA template used for cloning was synthesized using the Biotech M-MLV reverse transcription kit and the synthesized cDNA template was stored at –20 °C. The transmembrane segments and topology of proteins were predicted by TMHMM v2.0 (http://www.cbs.dtu.dk/services/TMHMM-2.0/) (*Krogh et al., 2001*; *Wu et al., 2017*). Multiple alignments of the complete amino acid sequences were performed with Clustal Omega (http://www.ebi.ac.uk/Tools/msa/clustalo).

## qRT-PCR for spatial expression pattern and RNAi efficiency

The nervous system (NS), fat body (Fb), digestive system (DS), epidermis (EP), male reproductive organs (MRO), and female reproductive organs (FRO) from 3 to 5 days old adults were dissected in 1x PBS, to determine the tissue-specific expression pattern. With the exception of the reproductive system, the rest of the tissue comes from both sexes equally. The adult BPHs were placed on ice under slight anesthesia and dissected under a stereoscopic microscope (ZEISS). At least 50 individuals were dissected as one sample for tissue-specific analysis with three replicates. All tissues were quickly transferred to Trizol (Invitrogen, Carlsbad, CA, USA) solution at 0 ° C after dissection.

For RNAi efficiency testing, we collected the whole bodies of BPHs and transferred to a 1.5 mL EP tube containing Trizol reagent. Total RNA extraction was using Trizol reagent (Invitrogen, Carlsbad, CA, USA) according to the manufacturer's instructions. The cDNA template used for cloning was synthesized using the Biotech M-MLV reverse transcription kit and the synthesized cDNA template was stored at –20 °C.

Real-time qPCRs of the various samples used the UltraSYBR Mixture (with ROX) Kit (CWBIO, Beijing, China). The PCR was performed in 20 µl reaction, including 2 µl of 10-fold diluted cDNA, 1 µl of each primer (10 µM), 10 µl 2×UltraSYBR Mixture, and 6 µl RNase-free water. The PCR conditions used were as follows: initial incubation at 95 °C for 10 min, followed by 40 cycles of 95 °C for 10 s and 60 °C for 45 s. *N. lugens* 18 S rRNA and actin were used as an internal control (primers listed in *Supplementary file 1*). Relative quantification was performed via the comparative $2^{-\Delta\Delta CT}$ method (*Krogh et al., 2001*; *Livak and Schmittgen, 2001*).

## RNAi and RNAi efficiency determination

For lab-synthesized dsRNA, *gfp*, Nl*CrzR,* and Nl*Crz* were amplified by PCR using specific primers conjugated with the T7 RNA polymerase promoter (primers listed in *Supplementary file 1*). dsRNA was synthesized by the MEGAscript T7 transcription kit (Ambion, Austin, TX, USA) according to the manufacturer's instructions. Finally, the quality and size of the dsRNA products were verified by 1% agarose gel electrophoresis and the Nanodrop 1000 spectrophotometer and kept at –70 °C until use.

3–5 days old virgin female BPHs were used for injection of 50 nl of 5 µg/µl dsRNA per insect. Injection of an equal volume of ds*gfp* was used as a negative control. After 48 hr, the insects were used behavior assays and gene relative expression analysis.

## Neuropeptide injection after RNAi

48 hr after the ds*CrzR* injection, each insect was injected again with 10 ng of peptide dissolved in 50 nL of 1x PBS balanced salt solution. Recovery was for 6 hr after injection. These insects were used for behavior assays. RNAi efficiency was examined by qPCR using a pool of 10 individuals 48 hr after dsRNA injections.

## In vitro synthesis of sgRNA and Cas9 mRNA

The single guide RNA (sgRNA) was designed as previously reported (*Sun et al., 2023a*; *Livak and Schmittgen, 2001*). Briefly, sgRNA was designed by manually searching genomic sequence around the region of the *Crz* and *CrzR* exon for the sequences corresponding to 5'-$N_{17-20}$NGG-3', where NGG is the protospacer-adjacent motif (PAM) of SpCas9 and N is any nucleotide. For in vitro transcription of sgRNA, were synthesized in vitro using the GeneArtPrecision gRNA Synthesis Kit (Thermo Fisher Scientific, Vilnius, Lithuania) according to the instructions of the manufacturer. The Cas9 protein (TrueCutCas9 Protein v2, Cat. NO. A36497) was purchased from Thermo Fisher Scientific (Shanghai, China).

## Embryo microinjection, crossing, and genotyping

The embryonic injection was performed as previously reported (*Zhang et al., 2023*; *Sun et al., 2023a*). The female BPHs, which had been mated for 3–6 days after emergence, were selected to lay eggs. After 30 min, the embryos were transferred onto double-sided tape attached to a microscope slide along the long edge by gently pressing the slide onto the dorsal surface of embryos. A few drops of halocarbon oil 700 was added to these eggs. About 0.5–1 nL mixture of Cas9 protein (200 ng/µL) and sgRNA (100 ng/µL) was injected into each egg using a FemtoJet and Inject Man NI 2 microinjection system (Eppendorf, Germany). After the injection, the eggs were placed in Petri dishes covered with moist filter paper, and the Petri dishes were placed in plant growth chamber at 27±1 °C, 70±10% RH for hatching. These eggs hatched into G0 generations.

## Crossing and genotyping

To establish homozygous mutants of *Nilaparvata lugens* through CRISPR-mediated gene editing, newly emerged G0 adults were outcrossed with wild-type counterparts to generate G1 progeny. Following successful mating and oviposition, genomic DNA was extracted from G0 adults for mutation screening. The target region surrounding NlCrzR was amplified by PCR using specific primers NlCrzR-check-F and NlCrzR-check-R (primer sequences listed in *Supplementary file 1*), followed by Sanger sequencing to identify G0 individuals carrying mutations.

Progeny (G1) from mutation-positive G0 parents were maintained through sibling crosses to establish G2 populations. Subsequent genotyping via Sanger sequencing was performed on G1 individuals to select breeding pairs where both male and female parents carried heterozygous mutations. The

resulting G2 offspring from these validated pairs were then subjected to intercrossing to ultimately generate homozygous mutant lines.

After homozygous mutant lines were established, virgin homozygous mutant females were paired individually with virgin wild-type males. Following successful copulation, inseminated females were transferred to fresh rice seedlings for oviposition, and fertilized eggs were allowed to hatch. Hatched nymphs were continuously reared on fresh rice seedlings until the fifth instar. Rice seedlings were replaced every 12 hr, and newly emerged adults were collected and sex-separated for subsequent behavioral assays.

## Quantitative RT-PCR

The first-strand cDNA was synthesized with HiScript II Q RT SuperMix for qPCR (+gDNA wiper) kit (Vazyme, Nanjing, China) using an oligo (dT)18 primer and 500 ng total RNA template in a 10 µl reaction, following the instructions. Real-time qPCRs in the various samples used the UltraSYBR Mixture (with ROX) Kit (CWBIO, Beijing, China). The PCR was performed in 20 µl reaction, including 4 µl of 10-fold diluted cDNA, 1 µl of each primer (10 µM), 10 µl 2xUltraSYBR Mixture, and 6 µl RNase-free water. The PCR conditions used were as follows: initial incubation at 95 °C for 10 min, followed by 40 cycles of 95 °C for 10 s and 60 °C for 45 s. *N. lugens* 18 S rRNA or *D. melanogaster* rp49 were used as an internal control (*Supplementary file 1*). Relative quantification was performed via the comparative $2^{-\Delta\Delta CT}$ method (*Livak and Schmittgen, 2001*).

## Immunohistochemistry

Adult BPHs, 3–5 days old, were dissected under 1 x phosphate-buffered saline (PBS; pH 7.4) in Schneider's insect medium (S2). We dissected the brain, ventral nerve cord, and reproductive system of the BPHs. These tissues were fixed in 4% paraformaldehyde in PBS for 30 min at room temperature. After extensive washing with PTX (0.5% Triton X100, 0.5% bovine serum albumin in PBS), blocked in 3% normal goat serum. Then, the tissues were incubated in Anti - CRZ for 12 hr at 4°C and in secondary antibody for 12 hr at 4°C. Primary antibodies used were: rabbit anti - CRZ (1:1000, Anti - CRZ was gift from Jan A. Veenstra), mouse anti-Bruchpilot (1:1000, Developmental Studies Hybridoma Bank nc82). Secondary antibodies used: donkey anti-rabbit IgG conjugated to Alexa 488 (1:500, R37118, Thermo Fisher Scientific) and donkey anti-mouse IgG conjugated to Alexa 555 (1:500, R37115, Thermo Fisher Scientific). The samples were mounted in Vectorshield (Vector Laboratory). Images were acquired with Zeiss LSM 700 confocal microscopes, and were processed with ImageJ software. All antibodies were diluted in PTX solution. The primary antibody was diluted at a ratio of 1:1000, while the secondary antibody was diluted at a ratio of 1:500.

Adult female *D. melanogaster*, 3–5 days old, were dissected under 1 x phosphate-buffered saline (PBS; pH 7.4) in Schneider's insect medium (S2). We dissected the brain, ventral nerve cord, and reproductive system. These tissues were fixed in 4% paraformaldehyde in PBS for 30 min at room temperature. After extensive washing with PTX (0.5% Triton X100, 0.5% bovine serum albumin in PBS), blocked in 3% normal goat serum. Then, the tissues were incubated in Anti - CRZ and Anti - nc82 for 12 hr at 4°C and in secondary antibody for 12 hr at 4°C. Primary antibodies used were: rabbit anti-CRZ (1:1000, A11122, Provided by Jan A. Veenstra). Secondary antibodies used: donkey anti-rabbit IgG conjugated to Alexa 488 and anti-mouse IgG conjugated to Alexa 555 (R37118, Thermo Fisher Scientific). The samples were mounted in Vectorshield (Vector Laboratory). Images were acquired with Zeiss LSM 700 confocal microscopes, and were processed with ImageJ software. All antibodies were diluted in PTX solution.

## Fluorescent in situ hybridization (FISH)

FISH was performed as previously reported by *Yan et al., 2022*; *Zhang et al., 2023*. Briefly, the reproductive system of the female BPH dissection was performed under 1 x Phosphate Buffered Saline (PBS) supplemented with a protease and phosphatase inhibitor cocktail (Shyuanye, #R40012) on a clean, clear rubber pad using two fine forceps (Dumont, #0108–5-po). Dissected tissues were fixed using 4% paraformaldehyde (PFA) (Shyuanye, #22039) diluted in 1 x PBS with 0.1% Tween20 (0.1% PBST) at room temperature (RT) for 20 min on a shaker. Fixed tissues were washed three times with 0.1% PBST for 15 min each on a shaker. The tissues were dehydrated in 25%, 50%, 90%, and 100% methanol, and then rehydrated in reverse order, performed at room temperature for 10 min at a time. The tissue was

then washed twice with 0.1% PBST and shaken for 5 min. A post fixation was performed by incubating the tissue in 4% PFA in 1 x PBS on a shaker for 20 min, followed by two washes with PBST for 5 min each while shaking. The tissue was permeabilized with proteinase K (10 μg/ml) at room temperature for 2 min. After incubation, the tissue was washed with 0.1% PBST for 5 min while shaking. The PBST was removed, and pre-hybridization solution (Boster, #AR0152) was applied at 55 °C for approximately 2 hr. The DIG probes were denatured in a metal bath at 80 °C for 5 min and then immediately placed on ice. The denatured DIG-labeled probes were added to the hybridization solution, and hybridization was carried out overnight at 55 °C. Then, the hybridized tissue was washed twice at 55 °C for 40 min with warm pre-hybridization solution, followed by four washes at room temperature with 0.1% PBST for 10 min each. The detection of the DIG-labeled probes was performed using an anti-DIG conjugated fluorescent antibody (1:100 dilution, Jackson, #200-542-156), incubated at room temperature in 0.1% PBST for 2 hr, followed by three washes with 0.1% PBST. Finally, the tissue was mounted in mounting medium and imaged using a Zeiss confocal microscope.

## Quantification and statistical analysis

All graphs were generated using Prism 9 software (GraphPad Software, La Jolla, CA). Data presented in this study were first verified for normal distribution by D'Agostino-Pearson normality test. If normally distributed, Student's $t$-test was used for pairwise comparisons, and one-way ANOVA and chi-square was used for comparisons among multiple groups. If not normally distributed, Mann-Whitney test was used for pairwise comparisons, and Kruskal-Wallis test was used for comparisons among multiple groups, followed by Dunn's multiple comparisons. All data are presented as mean ± s.e.m.

## Acknowledgements

We thank Dr. Jan A Veenstra for providing CRZ antibody. This project was supported by the National Key R&D Program of China (2022YFD1700200) to S-FW, the National Natural Science Foundation of China (No. 32472542) to S-FW.

## Additional information

### Funding

| Funder | Grant reference number | Author |
| --- | --- | --- |
| National Key Research and Development Program of China | 2022YFD1700200 | Shun-Fan Wu |
| National Natural Science Foundation of China | 32472542 | Shun-Fan Wu |

The funders had no role in study design, data collection and interpretation, or the decision to submit the work for publication.

### Author contributions

Ning Zhang, Conceptualization, Resources, Data curation, Software, Formal analysis, Validation, Investigation, Methodology, Writing – original draft; Shao-Cong Su, Formal analysis, Investigation, Methodology; Ruo-Tong Bu, Data curation, Software, Investigation, Methodology; Yijie Zhang, Data curation, Investigation; Lei Yang, Data curation, Validation, Investigation; Jie Chen, Formal analysis, Validation; Dick R Nässel, Conceptualization, Writing – review and editing; Congfen Gao, Resources, Visualization; Shun-Fan Wu, Conceptualization, Resources, Formal analysis, Supervision, Funding acquisition, Investigation, Methodology, Writing – original draft, Project administration, Writing – review and editing

### Author ORCIDs

Ning Zhang https://orcid.org/0009-0003-5402-5107
Shao-Cong Su https://orcid.org/0009-0007-3782-6657
Shun-Fan Wu https://orcid.org/0000-0003-0096-147X

Reviewer #2 (Public review): https://doi.org/10.7554/eLife.109297.3.sa1
Author response https://doi.org/10.7554/eLife.109297.3.sa2

## Additional files

### Supplementary files
MDAR checklist

Supplementary file 1. Primers used in this study.

### Data availability

All data generated or analyzed during this study are included in the manuscript, supporting files, and public repositories. Source data files for the relevant figures and tables are provided with the submission. Raw confocal imaging data underlying this study have been deposited in Dryad: DOI: https://doi.org/10.5061/dryad.qz612jmvj.

The following dataset was generated:

| Author(s) | Year | Dataset title | Dataset URL | Database and Identifier |
|---|---|---|---|---|
| Zhang N, Su S-C, Bu R-T, Zhang Y-J, Yang L, Chen J, Nässel DR, Gao C-F, Wu S-F | 2026 | Endogenous corazonin signaling modulates the post-mating switch in behavior and physiology in females of the brown planthopper and Drosophila | https://doi.org/10.5061/dryad.qz612jmvj | Dryad Digital Repository, 10.5061/dryad.qz612jmvj |

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
