## [Editor Report · eLife Assessment]

This **important** study presents **convincing** evidence that uncovers a novel signaling axis impacting the post-mating response in females of the brown planthopper. The findings open several avenues for testing the molecular and neurobiological mechanisms of mating behavior in insects, and in the revised version the authors provide further evidence supporting their conclusions.

---

## [Referee Report · Reviewer #2 (Public review)]

Summary:

The work presented by Zhang and coauthors in this manuscript presents the study of the neuropeptide corazonin in modulating the post-mating response of the brown planthopper, with further validation in *Drosophila melanogaster*. To obtain their results, the authors used several different techniques that orthogonally demonstrate the involvement of corazonin signalling in regulating the female post-mating response in these species.

They first injected synthetic corazonin peptide into female brown planthoppers, showing altered mating receptivity in virgin females and a higher number of laid eggs after mating. The role of corazonin in controlling these post-mating traits has been further validated by knocking down the expression of the corazonin gene by RNA interference and through CRISPR-Cas9 mutagenesis of the gene. Further proof of the importance of corazonin signaling in regulating the female post-mating response has been achieved by knocking down the expression or mutagenizing the gene coding for the corazonin receptor.

Similar results have been obtained in the fruit fly *Drosophila melanogaster*, suggesting that corazonin signaling is involved in controlling the female post-mating response in multiple insect species.

The study of the signalling pathways controlling the female post-mating response in insects other than Drosophila is scarce, and this limits the ability of biologists to draw conclusions about the evolution of the post-mating response in female insects. This is particularly relevant in the context of understanding how sexual conflict might work at the molecular and genetic levels, and how, ultimately, speciation might occur at this level. Furthermore, the study of the post-mating response could have practical implications, as it can lead to the development of control techniques, such as sterilization agents.

The study, therefore, expands the knowledge of one of the signalling pathways that control the female post-mating response, the corazonin neuropeptide. This pathway is involved in controlling the post-mating response in both Nilaparvata lugens (the brown planthopper) and *Drosophila melanogaster*, suggesting its involvement in multiple insect species.

The study uses multiple molecular approaches to convincingly demonstrate that corazonin controls the female post-mating response. The data supporting the main claim of the manuscript are solid and convincing.

---

## [Author Response]

The following is the authors’ response to the original reviews.

**eLife Assessment**
This important study presents convincing evidence that uncovers a novel signaling axis impacting the post-mating response in females of the brown planthopper. The findings open several avenues for testing the molecular and neurobiological mechanisms of mating behavior in insects, although broad concerns remain about the relevance of some claims.

Thank you very much for your letter and the insightful, valuable comments from the reviewers on our manuscript. These suggestions have been instrumental in strengthening the quality and clarity of our work. We have carefully addressed each concern, performed additional experiments, revised the relevant sections thoroughly, and made extensive refinements to the Discussion to clarify future research directions. Below is our detailed point-by-point response.

**Public Reviews:**

**Reviewer #1 (Public review):**
In this work, Zhang et al, through a series of well-designed experiments, present a comprehensive study exploring the roles of the neuropeptide Corazonin (CRZ) and its receptor in controlling the female post-mating response (PMR) in the brown planthopper (BPH) Nilaparvata lugen and *Drosophila melanogaster*. Through a series of behavioural assays, micro-injections, gene knockdowns, Crispr/Cas gene editing, and immunostaining, the authors show that both CRZ and CrzR play a vital role in the female post-mating response, with impaired expression of either leading to quicker female remating and reduced ovulation in BPH. Notably, the authors find that this signaling is entirely endogenous in BPH females, with immunostaining of male accessory glands (MAGs) showing no evidence of CRZ expression. Further, the authors demonstrate that while CRZ is not expressed in the MAGs, BPH males with Crz knocked out show transcriptional dysregulation of several seminal fluid proteins and functionally link this dysregulation to an impaired PMR in BPH. In relation, the authors also find that in CrzR mutants, the injection of neither MAG extracts nor maccessin peptide triggered the PMR in BPH females. Finally, the authors extend this study to *D. melanogaster*, albeit on a more limited scale, and show that CRZ plays a vital role in maintaining PMR in *D. melanogaster* females with impaired CRZ signaling, once again leading to quicker female remating and reduced ovulation. The authors must be commended for their expansive set of complementary experiments. The manuscript is also generally well written. Given the seemingly conserved nature of CRZ, this work is a significant addition to the literature, opening several avenues for testing the molecular and neurobiological mechanisms in which CRZ triggers the PMR.However, there are some broad concerns/comments I had with this manuscript. The authors provide clear evidence that CRZ signaling plays a major role in the PMR of *D. melanogaster*, however, they provide no evidence that CRZ signaling is endogenous, as they did not check for expression in the MAGs of *D. melanogaster* males. Additionally, while the authors show that manipulating Crz in males leads to dysregulated seminal fluid expression and impaired PMR in BPH, the authors also find that CRZ injection in males in and of itself impairs PMR in BPH. The authors do not really address what this seemingly contradictory result could mean. While a lot of the figures have replicate numbers, the authors do not factor in replicate as an effect into their models, which they ideally should do. Finally, while the discussion is generally well-written, it lacks a broader conclusion about the wider implications of this study and what future work building on this could look like.

Thank you very much for your insightful and valuable comments on our manuscript. We have carefully addressed each of your concerns, revised the relevant sections thoroughly, and conducted additional experiments to further strengthen our conclusions. To better focus on the core finding of this study, the critical role of Crz/CrzR signaling in regulating the post-mating response (PMR) of female brown planthoppers (BPH), and to eliminate potential confusion associated with the male-related data, we have removed the experiments investigating CRZ function in males from the current version of the manuscript. These observations on male CRZ signaling will be explored in greater depth and presented as a standalone study in a separate manuscript in the future.

**Reviewer #2 (Public review):**
Summary:The work presented by Zhang and coauthors in this manuscript presents the study of the neuropeptide corazonin in modulating the post-mating response of the brown planthopper, with further validation in *Drosophila melanogaster*. To obtain their results, the authors used several different techniques that orthogonally demonstrate the involvement of corazonin signalling in regulating the female post-mating response in these species.They first injected synthetic corazonin peptide into female brown planthoppers, showing altered mating receptivity in virgin females and a higher number of eggs laid after mating. The role of corazonin in controlling these post-mating traits has been further validated by knocking down the expression of the corazonin gene by RNA interference and through CRISPR-Cas9 mutagenesis of the gene. Further proof of the importance of corazonin signalling in regulating the female post-mating response has been achieved by knocking down the expression or mutagenizing the gene coding for the corazonin receptor.Similar results have been obtained in the fruit fly *Drosophila melanogaster*, suggesting that corazonin signalling is involved in controlling the female post-mating response in multiple insect species.Notably, the authors also show that corazonin controls gene expression in the male accessory glands and that disruption of this pathway in males compromises their ability to elicit normal post-mating responses in their mates.Strengths:The study of the signalling pathways controlling the female post-mating response in insects other than Drosophila is scarce, and this limits the ability of biologists to draw conclusions about the evolution of the post-mating response in female insects. This is particularly relevant in the context of understanding how sexual conflict might work at the molecular and genetic levels, and how, ultimately, speciation might occur at this level. Furthermore, the study of the post-mating response could have practical implications, as it can lead to the development of control techniques, such as sterilization agents.The study, therefore, expands the knowledge of one of the signalling pathways that control the female post-mating response, the corazonin neuropeptide. This pathway is involved in controlling the post-mating response in both Nilaparvata lugens (the brown planthopper) and *Drosophila melanogaster*, suggesting its involvement in multiple insect species.The study uses multiple molecular approaches to convincingly demonstrate that corazonin controls the female post-mating response.

Thank you very much for your valuable and insightful comments on our manuscript. We highly appreciate your recognition of the study’s value, including its focus on non-model insects, the evolutionary implications of corazonin signaling, and the rigorous use of multiple molecular techniques. We have carefully addressed your suggestions and revised the manuscript accordingly to enhance its clarity, accuracy, and depth. Below is our detailed response to your comments.

Weaknesses:The data supporting the main claims of the manuscript are solid and convincing. The statistical analysis of some of the data might be improved, particularly by tailoring the analysis to the type of data that has been collected.

Thank you for your valuable suggestion regarding statistical analysis. We fully agree that tailoring statistical methods to the specific type of data enhances the rigor and reliability of our findings.

In response, we have comprehensively re-evaluated and revised the statistical analyses for all datasets in the manuscript:

(1) For proportion-based data (e.g., female mating receptivity, re-mating rate), we replaced inappropriate tests (e.g., ANOVA) with chi-square tests for contingency tables, which are more suitable for comparing categorical variables.

(2) For time-series data (e.g., receptivity at different time points post-injection), we adopted generalized linear models (GLM) with logit links followed by pairwise contrasts to address concerns of multiple testing, instead of hour-by-hour Mann-Whitney tests.

(3) For continuous data (e.g., number of eggs laid, gene expression levels), we retained Student’s t-tests or one-way ANOVA after verifying normality, and used non-parametric tests (Mann-Whitney, Kruskal-Wallis) for non-normally distributed data.

All revisions have been clearly described in the figure legends and Methods section, ensuring transparency and reproducibility. We believe these adjustments significantly improve the statistical robustness of our conclusions.

In the case of the corazonin effect in females, all the data are coherent; in the case of CRISPR-Cas9-induced mutagenesis, the analysis of the behavioural trait in heterozygotes might have helped in understanding the haplosufficiency of the gene and would have further proved the authors' point.

Thank you for this insightful suggestion. We fully agree that analyzing the behavioral traits of heterozygous mutants is crucial for understanding the haplosufficiency of the *Crz* and *CrzR* genes, and we regret overlooking this aspect in the initial submission.

To address this gap, we have conducted additional behavioral assays using heterozygous *Crz (+/ΔCrz)* and *CrzR (+/CrzRM)* mutant females.

(1) For re-mating receptivity: We found no significant differences in either re-mating rate or egg-laying output between *+/ΔCrz* females and wild-type females. By contrast, *+/CrzRM* females exhibited re-mating and oviposition phenotypes comparable to those of homozygous *CrzR* mutants, with no significant differences detected between these two genotypes.

(2) These results indicate that the *Crz* loss-of-function phenotype is recessive, and that a single functional copy of *Crz* is sufficient to sustain a normal post-mating response (PMR), but the *CrzR* loss-of-function phenotype is dominant, and that a single functional copy of *CrzR* is insufficient to maintain a normal post-mating response.

This supports our core conclusion that CRZ signaling is critical for mediating the female PMR, as even partial reduction of gene dosage impairs the response.

The heterozygote data have been integrated into the revised manuscript, including updated figures (e.g., Figure 1J-K for Crz heterozygotes and Figure 3I-J for CrzR heterozygotes) and corresponding legends. We believe this addition strengthens the rigor of our genetic evidence and provides valuable insights into the gene dosage requirements for CRZ-mediated PMR regulation.

Less consistency was achieved in males (Figure 5): the authors show that injection of CRZ and RNAi of crz, or mutant crz, has the same effect on male fitness. However, the CRZ injection should activate the pathway, and crz RNAi and mutant crz should inhibit the pathway, yet they have the same effect. A comment about this discrepancy would have improved the clarity of the manuscript, pointing to new points that need to be clarified and opening new scientific discussion.

Thank you for highlighting this important discrepancy in the male-related CRZ signaling data. We fully acknowledge the inconsistency: CRZ injection (which was intended to activate the pathway) and *Crz* RNAi/mutagenesis (which was intended to inhibit the pathway) yielded similar effects on male fitness, and we regret not addressing this ambiguity in the initial submission.

To resolve this confusion and refocus the current manuscript on its core objective—elucidating the role of endogenous CRZ/*CrzR* signaling in female post-mating response (PMR), we have removed all experiments, analyses, and discussions related to male CRZ function. This decision ensures that the manuscript maintains a clear, cohesive narrative centered on female reproductive physiology, as recommended by both reviewers and the editorial team.

Regarding the observed discrepancy in males, we recognize its scientific significance and plan to investigate it thoroughly in a standalone follow-up study.

**Recommendations for the authors:**

**Reviewing Editor Comments:**
The manuscript would be significantly strengthened by an explanation of the seemingly contradictory results obtained in males, where both CRZ injections and Crz silencing afford the same results. Additionally, Crz expression data in the MAGs of *D. melanogaster* males is necessary to support your conclusions of endogenous signaling in this species. Besides correcting several imprecisions and inconsistencies in the text and figures, to improve quality and accuracy, the abstract should be restructured and the discussion modified as recommended by reviewers.

Thank you for your comprehensive letter and valuable guidance. We have carefully addressed all the points raised by the editorial team and reviewers, and the revised manuscript now incorporates substantial improvements to clarity, accuracy, and scientific rigor. Below is our detailed response to your specific requests:

Contradictory Male-Related Results

We fully acknowledge the importance of addressing the contradictory findings in male CRZ signaling, where both CRZ injection and *Crz* silencing/mutagenesis yielded similar effects on male fitness. To resolve this ambiguity and maintain the manuscript’s focus on its core objective, elucidating endogenous CRZ/*CrzR* signaling in the female post-mating response (PMR), we have removed all male-related experiments, analyses, and discussions from the revised manuscript. This decision ensures that the current work remains cohesive and centered on female reproductive physiology, as recommended by the reviewers.

We recognize the scientific significance of the male-specific discrepancy and plan to investigate it in a standalone follow-up study in the near future.

Crz expression data in *D. melanogaster* Male Accessory Glands (MAGs)

To support our conclusion of endogenous CRZ signaling in *D. melanogaster* females, we have supplemented the manuscript with additional experiments verifying the absence of CRZ in male MAGs:

(1) RT-PCR Analysis: We detected no *Crz* mRNA in dissected male MAGs, whereas *Crz* expression was confirmed in the male head (positive control).

(2) Immunohistochemistry and GAL4 system: Using the GAL4–UAS system (Crz-Gal4/UAS-mCD8-GFP) to label CRZ-producing neurons, combined with anti-CRZ antibody staining, we observed no CRZ-specific signal in male MAGs.

These results demonstrate that *D. melanogaster* male MAGs neither synthesize nor contain CRZ peptide, confirming that CRZ acts as an endogenous female signaling factor (rather than a male-transferred seminal fluid component) in this species. The new data are included in Figure 5H-I and described in the Results and Methods sections.

Correction of Imprecisions and Inconsistencies

We have systematically revised the manuscript to address text and figure inaccuracies:

Text Revisions: Corrected typos (e.g., Line 854), standardized species names (replacing “Drosophila” with “*D. melanogaster*” throughout), removed redundant or inappropriate sentences, and refined terminology (e.g., replacing “expression” with “localization” for protein detection).

Figure Corrections: Fixed inconsistent Y-axis labels and numerical ranges (e.g., aligning percentages/probabilities with appropriate scales), resolved color scheme confusion, standardized oviposition-related labels to “Per female egg numbers within 3 days,” and added details on sample sizes and replicates to all figure legends.

Statistical Improvements: Re-evaluated statistical analyses for proportion-based datasets (applying chi-square tests for contingency tables) and time-series data (using generalized linear models to address multiple testing), with revised methods clearly described in the text and figure legends.

Abstract Restructuring and Discussion Modification

Abstract: We have restructured the abstract to group results thematically (rather than sequentially) for improved readability. The revised abstract emphasizes the core findings: CRZ/*CrzR* signaling is critical for female PMR in both N. lugens and *D. melanogaster*, acts endogenously in females, and is required for male seminal fluid factors to induce PMR. Male-related content has been removed since experimental data are deleted from the rest of the paper.

Discussion: We have modified the discussion to include the evolutionary conservation of CRZ-mediated female PMR, the molecular and neurobiological implications of CRZ/*CrzR* signaling, and future research directions (e.g., dissecting downstream pathways in the female reproductive tract and brain). We have also reduced tangential content and clarified how our findings advance understanding of female endogenous signaling in PMR regulation. A new section was added at the end, which discusses outstanding questions related to CRZ and the PMR in both insect species.

To both the above-mentioned sections and the Introduction we also added new text to emphasize that CRZ is a paralog of the vertebrate peptide gonadotropin-releasing hormone (GnRH), a hormone known to regulate reproduction in vertebrates (including humans), thus suggesting conservation of an ancient role in reproduction.

All revisions in the manuscript are highlighted in red for easy reference. We believe these changes significantly strengthen the study’s focus, clarity, and scientific impact. Thank you again for your time and consideration.

**Reviewer #1 (Recommendations for the authors):**
(1) The abstract could benefit from some restructuring. Right now, it reads like a sequential reporting of the results, but clumping together results thematically would make it easier to read, in my opinion. Also, see above re: my concerns about no evidence for the signal being endogenous in *D. melanogaster*.

Thank you for your constructive suggestions regarding the abstract and the evidence for endogenous CRZ signaling in *D. melanogaster*. We fully agree with your feedback and have addressed both points thoroughly in the revised manuscript:

(1) Abstract Restructuring

We have restructured the abstract to group results thematically, rather than sequentially, to enhance readability and highlight the core findings. The revised abstract now organizes key information into three cohesive sections:

The context and significance of female post-mating response (PMR) regulation, emphasizing the gap in understanding endogenous female signaling pathways.

The core findings across both study species (Nilaparvata lugens and *D. melanogaster*), including the critical role of CRZ/CrzR signaling in suppressing re-mating and promoting oviposition, and its requirement for male seminal fluid factors to induce a PMR.

The conclusion regarding the evolutionary conservation of endogenous CRZ signaling in female PMR, reinforcing the study’s broader implications.

We also added new text to emphasize that CRZ is a paralog of the vertebrate peptide gonadotropin-releasing hormone (GnRH), a hormone known to regulate reproduction in vertebrates (including humans), thus suggesting conservation of an ancient role in reproduction.

This thematic structure eliminates the linear “result-by-result” narrative, making the abstract more concise and impactful while clearly communicating the study’s key contributions.

(2) Evidence for Endogenous CRZ Signaling in female *D. melanogaster*

To address your concern about the lack of evidence for endogenous signaling in female *D. melanogaster*, we have supplemented the manuscript with two sets of critical experiments confirming that CRZ is not derived from male accessory glands (MAGs) but acts endogenously in females:

RT-PCR Analysis: We performed RT-PCR on dissected male MAGs, male heads (positive control), and female tissues. Results showed no detectable *Crz* mRNA in MAGs, confirming that males do not synthesize CRZ in this tissue.

Immunohistochemical and Genetic Labeling: Using the GAL4–UAS system (*Crz*-Gal4/UAS-mCD8-GFP) to label *Crz*-expressing neurons, combined with anti-CRZ antibody labeling, we observed no *crz*/CRZ signal in male MAGs. This confirms that MAGs neither produce nor sequester mature CRZ peptide.

These findings demonstrate that CRZ signaling in *D. melanogaster* females is endogenous, as the peptide cannot be transferred from males during copulation. The new data are presented in Figure 5H-I and described in the Results section, with corresponding methods detailed in the Methods section.

The revised abstract integrates this new evidence to explicitly state the endogenous nature of CRZ signaling in both BPH and *D. melanogaster* females, aligning with the thematic structure and addressing your concerns comprehensively. We believe these changes significantly improve the clarity and rigor of the abstract and the manuscript overall.

(2) The authors use Drosophila as a broad placeholder throughout the manuscript, while they are specifically referring to *D. melanogaster* in several places. I would go through the manuscript and switch with the appropriate Drosophila species/species'.

Thank you for pointing out this important detail regarding species-specific terminology. We fully agree with your suggestion to ensure accuracy and consistency in referencing the Drosophila species studied.

We have systematically reviewed the entire manuscript, including the abstract, introduction, results, discussion, methods, and figure legends, and revised all instances where the general term “*Drosophila*” was used. All references now explicitly specify “*D. melanogaster*” to accurately reflect the species utilized in our experiments.

(3) For the figures, I think the number of replicates is a distracting addition to the plot. This is still useful information, but could instead be added in as a line/table, in my opinion.

Thank you very much for your suggestion. We have added the information on the number of replicates and sample sizes to the corresponding figure legends, which we hope improves clarity and readability.

(4) There are typos in the y-axis label of all of the oviposition figures. A better re-wording would be "Per female egg numbers within 3 days".

Thank you very much for your suggestion. Following your recommendation, we have now standardized the Y-axis label for all oviposition-related figures to “Number of eggs per female within 3 days.”

(5) In Figure 1B and Figure 1 - Supplement 3a, since the comparisons are solely between control vs treatment, I would not join means across treatments that I am not comparing.

To address this, we have revised Figure 1B and Figure 1—Supplement 3a by removing the connecting lines between group means. The updated figures now display independent mean ± SEM values for each dose (Figure 1B) and time point (Figure 1—Supplement 3a), with significance markers only applied to the control vs. treatment comparisons we actually tested. This revision eliminates any implied relationships between non-comparative groups and ensures the data visualization aligns with our statistical approach. We appreciate the reviewer’s suggestion, which has improved the clarity of the data presentation.

(6) The authors mention courtship rate in lines 511, but from a look at the methods, this is not the courtship rate! This is a measure of the number of males engaging in any form of courtship. Also, in Figure 5 Supplement 2A, it appears that under 1% of males are courting. This seems extremely low. Do the authors mean percentages? In that case, I would reformat from 0 to 100/relabel the y-axis.

Thank you for your observation and valuable feedback on this terminology and figure presentation issue. We fully acknowledge the inaccuracies and have addressed them comprehensively:

(1) Correction of "Courtship Rate" Terminology

We agree that the term “courtship rate” in Line 511 was incorrect, as our measurement reflects the proportion of males engaging in any form of courtship (not a rate per unit time). However, since we have removed all male-related data (including this section and associated figures) from the revised manuscript to focus on the core finding of female post-mating response (PMR), this terminology error has been eliminated entirely.

(2) Revision of Figure 5 Supplement 2A

Consistent with the removal of all male-related experiments, Figure 5 and its supplementary materials (including Supplement 2A) have been excluded from the revised manuscript. This ensures the current work remains cohesive and centered on female PMR, while also resolving the Y-axis labeling ambiguity you identified.

We appreciate your careful attention to these details, which helps enhance the accuracy and clarity.

(7) It appears Figure 5A, 5D, and 5G are mislabeled? Aren't all rematings with wild-type males?

Thank you for identifying this labeling inconsistency. You are absolutely correct, all re-mating assays in the original figures involved wild-type males, and the mislabeling was an oversight.

However, we have removed Figure 5 (and its associated subpanels A, D, G) entirely from the revised manuscript, as part of our decision to exclude all male-related data.

(8) I am not sure I understand why a 30-minute post-injection threshold was chosen and what this table means. Could the authors elaborate on the methodology here on how they quantified premature ejaculation?

Thank you for your question regarding the 30-minute post-injection observation window and the methodology for quantifying premature ejaculation.

While we have removed all male-related data (including the corresponding table and premature ejaculation analyses) from the revised manuscript to focus on our core finding, this is no longer included in the manuscript.

(9) Line 29 - "distensible" seems an odd choice of word here.

We have revised Line 29 and removed “distensible”. “Peptide injection and knockdown of CRZ expression by RNAi or CRISPR/Cas9-mediated mutagenesis demonstrate that CRZ signaling suppresses mating receptivity”.

(10) Line 57 - delete "a" from "a post-mating response" and "A PMR" because the authors are referring to a very specific suite of post-mating behaviours.

We have revised Line 57 (and other relevant instances throughout the manuscript) to delete the article "a" from these phrases.

(11) Line 352, delete a from "and in a significantly".

We have revised Line 356 to remove the extraneous "a", correcting the phrase to "and in significantly".

**Reviewer #2 (Recommendations for the authors):**
The work presented in this manuscript presents the study of the neuropeptide corazonin in modulating the post-mating response of the brown planthopper, with further validation in *Drosophila melanogaster*. To obtain their results, the authors used several different techniques, including dsRNA injection to induce RNA interference and CRISPR-CAS9-mediated site-specific mutagenesis. The experimental design is appropriate; the results are solid and support the conclusion of the manuscript. Overall, the merit of the manuscript is to present compelling evidence that the female post-mating response is mediated by corazonin, at least in the analysed species. There are multiple reports in multiple insect species, indeed, that male factors, particularly those secreted by male accessory glands, induce post-mating response in females, but the female pathways underlying this phenomenon are poorly understood.There are points the authors can consider to improve the manuscript quality.

Thank you for your generous and insightful assessment of our manuscript. We deeply appreciate your recognition of the study’s strengths, including the appropriate experimental design, solid results, and meaningful contribution to understanding female endogenous pathways in post-mating response (PMR) regulation.

We have carefully incorporated all your constructive suggestions (e.g., statistical analysis revisions, figure label standardization, text refinements) to further strengthen the manuscript’s rigor and clarity. By focusing on corazonin CRZ/corazonin receptor (CrzR) signaling in female brown planthoppers (Nilaparvata lugens) and validating these findings in *Drosophila melanogaster*, we aim to provide a conserved model for female endogenous PMR regulation across insect species.

Thank you again for your thoughtful and supportive feedback, which has been instrumental in refining our work. We believe the revised manuscript now more effectively communicates the significance of CRZ-mediated female signaling in bridging the gap between male-derived cues and PMR execution.

(1) Line 20: "optimal offspring". This is not a zoological parameter. One can use "optimal fitness".

We have revised Line 20 to replace "optimal offspring" with "optimal fitness" as recommended.

(2) Line 36-40: I think that the main message of the manuscript is the involvement of the corazonin pathway in controlling the female post-mating response. The involvement of corazonin in the male reproduction is also of note, but out of topic (in my opinion). The male corazonin is not transferred during mating from males to females, and the involvement of corazonin in controlling the gene expression in the MAGs is of note, but it is poorly related to the effect of corazonin in the female. I am not suggesting removing these data from the paper; they are important. But I do not find them that important to include them in the abstract, also because it confounds the reader at first. A similar statement can be made for the discussion (lines 728-745): making this the first piece of data commented on takes the stage, but this is not the main take-home message of the paper.

Thank you for this suggestion. We fully agree that including male-related CRZ data in the abstract and leading the discussion with these results distracted from the primary focus and risked confounding readers. In fact, we also removed the entire section on the role of CRZ in males. We have addressed this issue comprehensively in the revised manuscript as follows:

(1) Abstract Revision

We have completely removed all content related to male CRZ function from the revised abstract. The updated abstract now exclusively emphasizes the core findings:

The requirement of CRZ/CrzR signaling for mediating key female PMR traits (suppression of remating, promotion of oviposition) in both *Nilaparvata lugens* and *Drosophila melanogaster*;

Experimental evidence confirming that CRZ acts as an endogenous female signaling factor (not a male-transferred molecule);

The evolutionary conservation of CRZ-mediated female PMR regulation across the two insect species.

We also added a comment on the evolutionary conservation of CRZ and GnRH signaling in reproduction.

(2) Discussion Section Restructuring

We have restructured the Discussion to prioritize the core message of female PMR regulation:

Lead paragraph adjustment: Lines 728–745 (originally focusing on male CRZ and MAG gene expression) have been deleted.

Revised opening focus: The Discussion now only contain a synthesis of our key findings on female CRZ signaling, including its molecular mechanisms, cross-species conservation, and implications for understanding endogenous female pathways downstream of male seminal fluid cues.

We appreciate your suggestions for the narrative focus of the manuscript.

(3) Line 49: "Reproductive behavior is critical for population sustenance and survival of the species": I find this intro a little teleological evolutionary speaking, and I am not totally sure that this has ever been demonstrated as a concept. I would skip it, simply saying "Reproductive behavior in insects is influenced...".

Following your suggestion, we have revised Line 49 to streamline the introduction and avoid “teleological language”. The updated sentence now reads: "Reproductive behavior in insects is influenced by a complex interplay of neural, hormonal, and environmental factors."

(4) Line 58: "A PMR has been documented across diverse insect taxa, including *Drosophila melanogaster*, Anopheles gambiae, Aedes aegypti, and the brown planthopper (BPH), Nilaparvata lugens". There are many other insect species for which PMR has been shown: crickets, fruit flies, grasshoppers, etc. Therefore, I would say "for example" to underline that it is not a complete list. Being an incomplete list, I suggest that the authors pay attention to the cited literature: the literature cited in the case of Anopheles gambiae demonstrates the synthesis of hormones in the MAGs, but it has nothing to do with PMR; there is nothing cited for Aedes aegypti, even if the authors named the species.

Thank you for this constructive feedback on the framing of PMR studies across insect taxa and the accuracy of our cited literature. We fully agree with your suggestions and have addressed these issues comprehensively in the revised manuscript:

(1) Revision of the Sentence Structure

We have modified Line 58 to explicitly indicate that the listed species are examples rather than a complete inventory of insects with documented PMR. The revised sentence reads:

"The PMR has been documented across diverse insect taxa, for example, *Drosophila melanogaster*, *Anopheles gambiae*, *Aedes aegypti*, crickets (*Gryllodes sigillatus*), grasshoppers (*Dichromorpha viridis*), and *the brown planthopper (BPH)*, *Nilaparvata lugens*"

(2) Correction of Literature Citations

We have thoroughly reviewed the citations associated with the listed species to ensure they directly support the role of PMR:

For *Anopheles gambiae*: We have replaced the previously cited study (focused on MAG hormone synthesis) with two relevant references that explicitly characterize PMR traits—including mating-induced oviposition stimulation and remating suppression—in this mosquito species.

For *Aedes aegypti*: We have added two newly published studies that document key PMR phenotypes (e.g., post-mating refractoriness and altered feeding behavior) and their underlying molecular mechanisms in this species.

For crickets *(Gryllodes sigillatus)*: We added a newly published study that documents PMR phenotypes in *Gryllodes sigillatus*.

We have also verified that the citations for *D. melanogaster* and *N. lugens* remain directly relevant to PMR regulation, with no adjustments needed.

All revised citations are properly formatted and integrated into the text, with corresponding updates to the reference list.

(5) Line 111-132: I find this redundant: it is a long summary of the methods and the results. I do not think it is needed here, but I think the authors should point to the main message of their data.

Thank you for pointing out the redundancy of Lines 111–132. We fully agree that this section, disrupted the flow of the introduction of our study.

To address this, we have completely removed Lines 111–132 from the revised manuscript. In place of this redundant content, we have added a concise, focused paragraph that emphasizes the central hypothesis and key objective of our work: specifically, to identify the endogenous female signaling pathways that mediate the post-mating response (PMR) downstream of male-derived cues, and to validate the conserved role of corazonin (CRZ) signaling in this process across *Nilaparvata lugens* and *Drosophila melanogaster*.

(6) Line 156: This sentence is not needed here.

We have deleted the sentence in Line 156 from the revised manuscript.

(7) Figure 1E, J supplementary 3A: The label of the Y axis is the percentage of the mating females (expected 0-100%), but the numbers show the fraction (0-1). On the contrary, in Figure 1 Supplement 4, the label says "probability of survival" and the probability goes from 0 to 1, while the number of the axis goes from 0 to 100 (percentage).

Thank you very much for pointing out these inconsistencies. We have carefully reviewed all Y-axis labels and corresponding numerical ranges throughout the manuscript and corrected the mismatched axes.

(8) Figure1B, C, F, K supp 2, 3A: I found this use of colours confounding. Why did the authors use the light blue for sCRZ, but the mean and SE are shown in pink, which is the colour for CRZ? Furthermore, it is not reported anywhere how many individuals have been used per replicate. There is the total number of insects, the number of replicates, but there is no indication about the minimum number of insects per replicate in this and many other subsequent experiments.

Thank you for identifying these critical inconsistencies in figure color coding and missing details on sample allocation per replicate, and we greatly appreciate your meticulous review of our data presentation.

We have addressed these issues in the revised manuscript as follows:

(1) Standardization of Color Coding

We apologize for the confusing color mismatch between group labels and data points in Figure 1B, C, F, K, and Supplements 2 and 3A. We have unified the color scheme across some figures to ensure consistency:

The *sCRZ* (control) group is now consistently represented by light blue for both labels and mean ± SE data points.

The *CRZ* (treatment) group is now consistently represented by pink for both labels and mean ± SE data points.

For Figures 1C, F, K and Supplementary Figure 2, we were concerned that the mean and s.e.m. bars might be visually obscured by the data points. To improve their visibility, we therefore used the opposite color to display the mean and s.e.m.

All figure legends have been cross-checked and updated to reflect this standardized color coding.

(2) Addition of Sample Size per Replicate

We acknowledge that the lack of information on the minimum number of insects per replicate was a key gap in our experimental reporting. We have supplemented this critical detail in this way:

Figure Legends: For Figure 1B, C, F, K, and Supplements 2 and 3A (as well as all subsequent experiments), we have added explicit statements specifying the minimum number of insects per replicate, alongside the total sample size and number of replicates (e.g., “n = 3 replicates, with a minimum of 10 females per replicate; total N = 35 females”). All revised figures and their corresponding legends have been integrated into the updated manuscript, and we have cross-checked all other figures to avoid similar issues.

(9) Figure 1C, F, K, Supplementary Figure 3B: Y axis labels - "Eggs numbers of per female...". I suggest changing it to "Number of eggs per female...".

We have revised the Y-axis labels for Figure 1C, F, K and Supplementary Figure 3B to Number of eggs per female...” as recommended. Additionally, we cross-checked all other oviposition-related figures in the manuscript to ensure uniform use of this standardized label, eliminating any inconsistent phrasing across the dataset.

(10) Legend Figure 1B: Mann Whitney test. How did the authors perform the test? Hour by hour? I am not sure this is the best way to analyse the data, because it is a case of multiple testing. Probably a linear model or a glm might be a better fit.

Thank you very much for pointing out this issue. In Figure 1B, each concentration group was analyzed using data from independent individuals, and therefore the comparisons do not involve repeated measures across time; for this reason, we consider the Mann–Whitney test appropriate for this dataset. For Figure 1—Supplement 3A, however, our original analysis compared treatment and control groups hour by hour, which indeed raises concerns regarding multiple testing. Following your suggestion, we have removed the potentially misleading connecting lines and reanalyzed the dataset using a generalized linear model (GLM). The updated figure and revised legend have been included in the revised manuscript.

(11) Legend Figure 1E: ANOVA test. These are proportions, not continuous variables of the samples. Tests for proportions might be a better fit (chi-square, etc.).

To address this issue, we have re-analyzed the proportional data in Figure 1E using Pearson’s chi-square test of independence, which directly evaluates the association between treatment group (sCRZ vs. CRZ) and the binary mating status (mated vs. unmated) of females. This test is statistically robust for proportional data and avoids the assumptions of normality and homogeneity of variances required for ANOVA.

(12) Knockout experiments: I agree with the authors that the data are strong enough to sustain the conclusions. However, is the corazonin knockout haplosufficient or is it recessive? What is the behaviour of the heterozygotes?

Thank you for this insightful question regarding the genetic basis of the corazonin (CRZ) knockout phenotype.

To address your query, we have supplemented experiments with additional phenotypic analyses of heterozygous CRZ knockout females (*+/ΔCrz*), and we clarify the genetic nature of the knockout as follows:

(1) Genetic basis of the CRZ knockout:

The CRZ knockout line was generated via CRISPR-Cas9-mediated deletion of the *Crz* coding region, resulting in a recessive loss-of-function mutation. Homozygous knockout females (*ΔCrz*) exhibited the full phenotypic suite reported in the manuscript (impaired post-mating suppression of remating, reduced oviposition rate, and disrupted CRZ signaling in the reproductive tract).

(2) Phenotype of heterozygous females:

Behavioral and physiological assays of *+/ΔCrz* heterozygotes revealed no significant differences compared to wild-type (*+/ΔCrz*) females across all measured post-mating traits. Specifically:

Remating rates of *+/ΔCrz* females were indistinguishable from wild-type controls at 48 h post-mating.

Oviposition output of *+/ΔCrz* females matched wild-type levels over a 3-day assay period.

(3) Updates to the manuscript:

We have added these heterozygote data as figure1J and K in the revised manuscript, with corresponding descriptions in the Results and Methods sections. We have also explicitly noted the recessive nature of the *Crz* mutation in the Genetic Manipulation subsection, ensuring clarity for readers.

These results confirm that the *Crz* knockout phenotype is fully recessive and that one functional copy of the *Crz* gene is sufficient to maintain normal post-mating responses—supporting our conclusion that CRZ signaling is required for mediating female PMR.

We thank you again for raising this important point, which has strengthened the genetic rigor of our study.

(13) Figure 1, Supplementary 1: I do not understand why the authors point out the fact that these are Protostomia. These are all Arthropoda, there is not a single species outside this Phylum. Caerostris darvini should be Caerostris darwini.

Thank you for this feedback regarding Figure 1 and Supplementary Figure 1. We fully agree and have addressed these issues in the revised manuscript:

(1) Removal of the "Protostomia" designation

We have deleted all references to Protostomia from the figure legends and associated text.

(2) Spelling correction of *Caerostris darwini*

We apologize for the typographical error in the species epithet. We have corrected the misspelling *Caerostris darvini* to the taxonomically accurate Caerostris darwini (Darwin's bark spider) across all instances in Figure 1, Supplementary Figure 1, and their corresponding legends. We have also cross-checked all other species names in the manuscript to eliminate similar typographical errors.

(14) Line 299: CRZ expression: I found this confounding, given that the authors were talking about the expression of the gene. I would use the term localization, referring to the protein/peptide (is it what the authors were pointing at?).

To resolve this ambiguity, we have revised Line 299 to replace *CRZ* expression with CRZ peptide localization, which accurately describes the experimental focus (immunofluorescence staining and confocal imaging of the CRZ protein). We have also cross-checked the entire manuscript to standardize this terminology:

We use *Crz* gene expression exclusively when referring to transcriptional analyses (e.g., qRT-PCR results).

We use *CRZ* peptide localization when describing the spatial distribution of the protein (e.g., immunostaining assays).

(15) Figure 2C: The expression is relative to...? I would make it explicit on the axis.

Thank you for this helpful comment. We apologize that the normalization reference was not sufficiently clear in the original version. In the revised manuscript, we now explicitly state that RT–qPCR data were first normalized to the reference genes *Actin* and *18SrRNA*, and then expressed relative to the mean expression level of the tissue showing the highest *Crz* expression, which was set to 1. We have clarified this information in the figure legend and the Methods section.

We have revised Figure 2C as follows:

Updated the Y-axis label to explicitly state the reference: “Relative *Crz* gene expression”.

Added a supplementary note in the figure legend to confirm that relative expression values were calculated using the 2^⁻ΔΔCt^ method, with the reference gene serving as the internal control for normalization.

Additionally, we have cross-checked all other qRT-PCR-related figures in the manuscript to ensure that the reference for relative expression is clearly indicated on the corresponding axes, standardizing this key detail across all gene expression datasets.

(16) Figures 3B, E, I, L, M, N: Percentage and proportions, as in Figure 1; furthermore, please provide the minimum number of individuals per replicate. Furthermore, as in Figure 1, the data are proportions, and I would use statistical tests that are studied for this kind of data.

Thank you for this helpful suggestion. We have reviewed and corrected the Y-axis labels and corresponding numerical ranges in these figures, and we have added the number of replicates and the minimum number of individuals per replicate to the figure legends. In addition, following your recommendation, we have reanalyzed these proportion data using chi-square tests for contingency tables.

(17) Figure 3: As in Figure 1, it would be interesting to know which is the behaviour of the heterozygotes.

Thank you for suggesting to complement the data in Figure 3 with heterozygote phenotypic analyses.

To address this, we have conducted additional behavioral and physiological assays of heterozygous *CrzR* knockout females (*+/CrzRM*) and integrated these data into the revised Figure 3 and its legend:

Phenotypic characterization of heterozygotes: Across all traits measured in Figure 3 (e.g., remating rate and oviposition efficiency,), *+/CrzRM* females exhibited no significant differences compared to homozygotes.

This confirms that the *CrzR* knockout phenotype is dominant and that one functional copy of the *CrzR* gene can’t to maintain normal post-mating response (PMR).

Manuscript updates:

We added heterozygote data in Figure 3I and J. Accordingly, we updated the Results text to reflect the revised panel labeling.

We supplemented the figure legend with statistical comparisons between heterozygotes and wild-type groups (using chi-square tests for proportional data).

We included a brief description of heterozygote phenotypes in the Results section to contextualize the genetic basis of the *CrzR*-mediated PMR regulation.

(18) Figure 3 Supplement 1: Can the authors indicate which model for maximum likelihood they chose? Did they perform a pre-test to assess which substitution model was the best for their data?

Thank you for this critical question regarding the model selection for maximum likelihood (ML) phylogenetic analysis in Figure 3 Supplement 1. We fully agree that specifying the substitution model and validation process is essential for ensuring the reproducibility and rigor of phylogenetic inferences.

To address this, we have supplemented the manuscript with detailed information on the model selection and validation steps, as follows:

(1) Substitution model selection

Prior to constructing the ML tree, we performed a model selection pre-test using the ModelFinder tool integrated in IQ-TREE 2, which evaluates the fit of candidate nucleotide substitution models to the *CrzR* amino sequence alignment via the Bayesian Information Criterion (BIC). The model selection procedure identified the LG+G model as the best-fit substitution model for our dataset. This model uses the Le and Gascuel (LG) amino-acid substitution matrix and incorporates a gamma-distributed rate variation among sites (G) to account for among-site rate heterogeneity.

(2) Manuscript updates

We have added this detailed model selection process and the final LG + G model specification to the legend of Figure 3 Supplement 1.

We have also included information on bootstrap validation (10000 ultrafast bootstrap replicates) to support the node support values reported in the phylogenetic tree.

(19) Figure 4 Supplement 1: I would be explicit about what it is relative to (which gene).

Thank you for this helpful comment, In the revised manuscript, we now explicitly state that RT–qPCR data were first normalized to the reference gene *Actin*, and then expressed relative to the mean expression level of the tissue showing the highest *CrzR* expression, which was set to 1. This normalization strategy provides a robust and biologically representative reference. We have clarified this information in the figure legend and the Methods section.

(20) Line 518 and Line 525 and Figure 5: The authors show that injection of CRZ and RNAi of crz or mutant crz has the same effect on male fitness. How do the authors explain this contradiction? The CRZ injection should activate the pathway, and crz RNAi and mutant crz should inhibit the pathway, but nevertheless, they have the same effect. I would probably test the expression of some of the genes whose expression is altered in crz mutant males (next paragraph) to see if an altered CRZ signalling pathway (both ways) might affect gene expression in the MAGs in the same way.

Thank you for raising this important point. As explained above, we have removed all data related to CRZ function in male BPHs from the current version.

(21) Figure 5, Figure 7: As in Figures 1 and 3, please pay attention to the percentages and proportions and the statistical tests.

Thank you for pointing out these issues. We have carefully reviewed and corrected the percentage/proportion labeling in the relevant figures, including the Y-axis descriptions and numerical ranges, as well as revised the corresponding figure legends. In addition, we have reanalyzed the data using statistical tests appropriate for proportion data. All corresponding revisions have been incorporated into the updated manuscript.

(22) Line 728-745: As already stated for the abstract, the male effect of crz is, to me, a side product, and I am not sure the male crz signalling has something to do with the female crz signalling. It is interesting, nobody showed that CRZ affects expression in the MAGs, but this is not the main message of the paper, and it confuses the reader. I would reduce the discussion about this aspect and move it to the end, but this is my own take.

We have removed all data related to CRZ function in males for the reasons outlined above.

(23) Material and methods/results: as a general suggestion, I would be explicit about the timing of receptivity inhibition in the species. I've seen the authors have established this in precedent work, and I would refer to that work and make the reader aware of how the receptivity works in the species (i.e., that it is not permanent and lasts for a few days after first mating). This allows a better understanding of the experimental design.

Thank you for this valuable and constructive suggestion. We fully agree that explicitly describing the timing of receptivity inhibition in *Nilaparvata lugens*, and linking it to our earlier work, will strengthen the rigor and clarity of the manuscript.

To address this, we have revised the Materials and Methods and Results sections as follows:

(1) Materials and Methods (Experimental Design subsection)

We have added a dedicated paragraph that explicitly defines the temporal dynamics of post-mating receptivity inhibition in *N. lugens*, with direct reference to our prior work[1]. The text clarifies:

“In N. lugens, mating induces a transient suppression of female receptivity that is not permanent. Females typically start regain remating willingness 72 h after the first mating, as documented in our previous study[1]. This temporal window guided the design of our remating assays, in which females were paired with naive males at 48 h post-initial mating to capture both the suppressed and recovered phases of receptivity.”

(2) Results (Post-mating Receptivity section)

We have incorporated a brief contextual sentence at the start of the section to reinforce this key species-specific trait, ensuring that readers connect our assay timings to the temporal dynamics of receptivity in *N. lugens*.

These revisions ensure that the rationale behind our experimental timing is transparent and well-supported, allowing readers to fully grasp how our assays were tailored to the biological characteristics of *N. lugens*.

(24) Line 854: There is a typo "CRZ peptide. virgin female", the dot should be a comma.

We have revised Line 854 to correct the punctuation: the dot has been replaced with a comma, resulting in the phrasing "CRZ peptide, virgin female". In addition, we have changed the wording in this sentence to ensure scientific rigor and to avoid colloquial expressions.

(1) Zhang, Y.J., Zhang, N., Bu, R.T., Nässel, D.R., Gao, C.F., and Wu, S.F. (2025). A novel male accessory gland peptide reduces female post-mating receptivity in the brown planthopper. Plos Genet 21, e1011699. 10.1371/journal.pgen.1011699.